# TET activity safeguards pluripotency throughout embryonic dormancy

Maximilian Stötzel[1,2], Chieh-Yu Cheng[1,2], Ibrahim A. Ilik[3], Abhishek Sampath Kumar [2,4], Persia Akbari Omgba[1,5], Vera A. van der Weijden [1], Yufei Zhang[6], Martin Vingron [6], Alexander Meissner [4], Tuğçe Aktaş [3], Helene Kretzmer[4] & Aydan Bulut-Karslıoğlu [1]✉

Dormancy is an essential biological process for the propagation of many life forms through generations and stressful conditions. Early embryos of many mammals are preservable for weeks to months within the uterus in a dormant state called diapause, which can be induced in vitro through mTOR inhibition. Cellular strategies that safeguard original cell identity within the silent genomic landscape of dormancy are not known. Here we show that the protection of *cis*-regulatory elements from silencing is key to maintaining pluripotency in the dormant state. We reveal a TET–transcription factor axis, in which TET-mediated DNA demethylation and recruitment of methylation-sensitive transcription factor TFE3 drive transcriptionally inert chromatin adaptations during dormancy transition. Perturbation of TET activity compromises pluripotency and survival of mouse embryos under dormancy, whereas its enhancement improves survival rates. Our results reveal an essential mechanism for propagating the cellular identity of dormant cells, with implications for regeneration and disease.

Dormancy equips organisms with a means to preserve cells over long periods of time to propagate species, to regenerate tissues or to overcome stressful conditions in the form of environmental insults or nutrient scarcity. Embryonic development of over 100 mammalian species features a safe pausing point at the blastocyst stage to adjust the timing of birth. This phenomenon, known as embryonic diapause, allows the preservation of the embryo for weeks to months in a reversibly dormant state[1]. Establishment of dormancy entails low anabolic activity and repression of transcriptional and translational programs[2–6]. A key feature of dormancy is the ability to revert back to proliferation without compromising the developmental potential and cell fate. Key pluripotency pathways such as LIF/STAT3 and WNT, which are dispensable in mouse preimplantation development yet stabilize the pluripotency of embryonic stem (ES) cells, are required

to also maintain pluripotency in diapause[7,8]. Yet, how pluripotency is preserved in the repressed genomic context of diapause is not known. Part of the challenge is the limited embryonic material that hinders dynamic perturbation studies. A diapause-like response can be triggered in mouse blastocysts and ES cells by direct inhibition of the master regulator of cell growth, mTOR (mTORi)[9]. mTORi-treated 'paused' ES cells transcriptionally and metabolically mimic in vivo-diapaused embryos and can be maintained in a near-dormant state for weeks to months without compromising pluripotency, thus providing an accessible model for mechanistic studies.

The uncommitted nature and broad developmental potential of pluripotent cells builds on the balance between a generally transcriptionally permissive genome and focal repression of developmental genes[10]. This highly transcribed open chromatin landscape powers rapid

[1]Stem Cell Chromatin Lab, Max Planck Institute for Molecular Genetics, Berlin, Germany. [2]Institute of Chemistry and Biochemistry, Department of Biology, Chemistry and Pharmacy, Freie Universität Berlin, Berlin, Germany. [3]Otto Warburg Laboratories, Max Planck Institute for Molecular Genetics, Berlin, Germany. [4]Department of Genome Regulation, Max Planck Institute for Molecular Genetics, Berlin, Germany. [5]Department of Mathematics and Computer Science, Freie Universität Berlin, Berlin, Germany. [6]Department of Computational Molecular Biology, Max Planck Institute for Molecular Genetics, Berlin, Germany. ✉e-mail: aydan.karslioglu@molgen.mpg.de

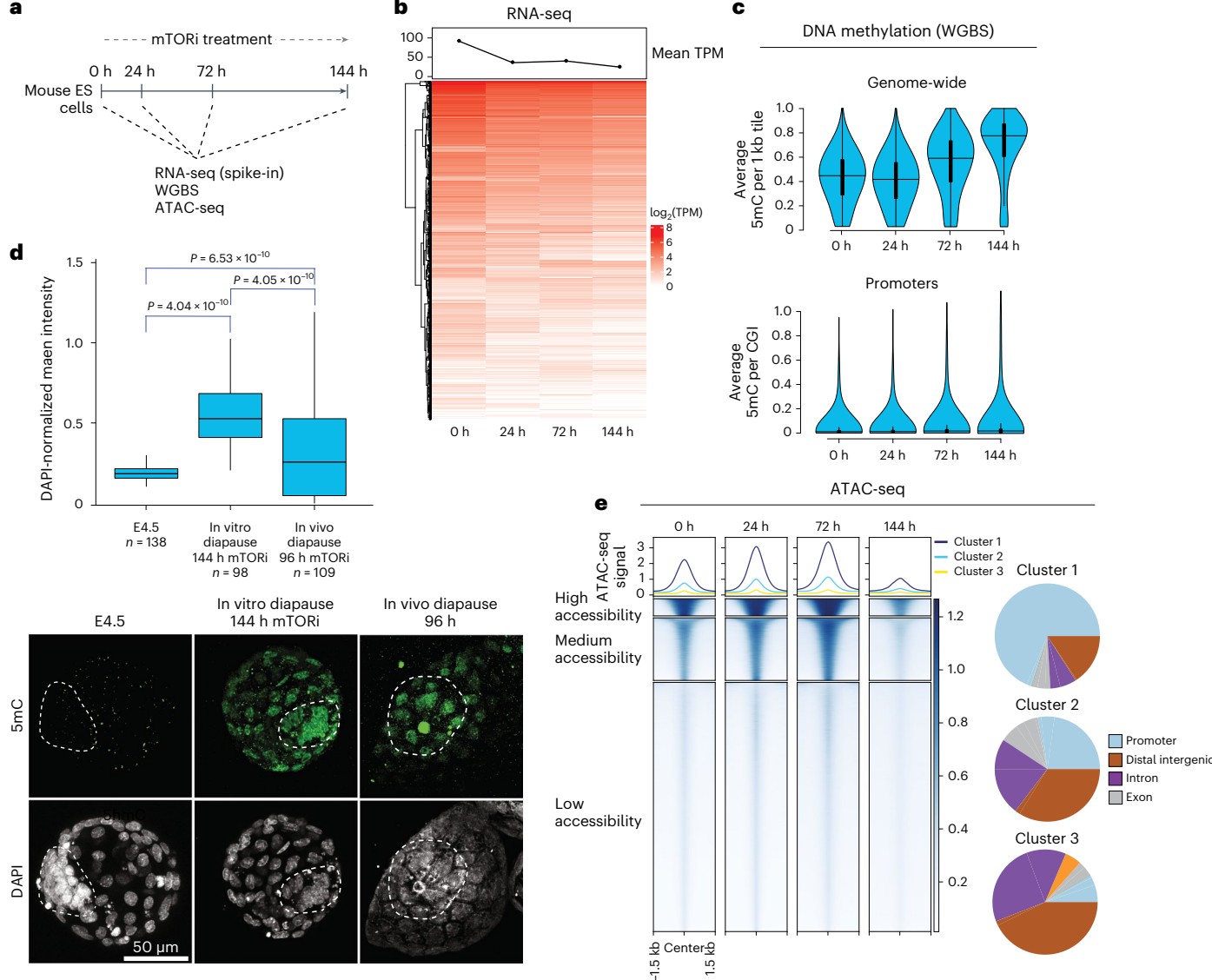

**Fig. 1 | Time-resolved genomic analysis of the transition of ES cells into dormancy. a**, Experimental workflow to profile transcriptional and chromatin features of ES cells entering mTORi-induced dormancy. Two replicates were performed for all experiments. Sequencing depth and quality control parameters can be found in Extended Data Fig. 1a. **b**, Bulk RNA-seq heatmap showing expression of all genes over time in ES cells treated with mTORi. All samples are normalized to ERCC[68] spike-in RNAs to accurately reflect global changes. The line plot on top shows mean TPM at each time point. **c**, DNA methylation levels in ES cells as mapped by whole genome bisulfite sequencing (WGBS). Top: average DNA methylation levels of the entire genome in 1 kb tiles. Bottom: DNA methylation levels of CpG islands (CGI). Horizontal lines show the median; vertical box plots within violin plots show interquartile range (IQR), and the whiskers show 1.5 IQR. **d**, Bottom: IF of E4.5, in vitro and in vivo-diapaused mouse blastocysts for 5mC methylation. Top: single-nucleus quantifications of mean 5mC intensity, normalized to DAPI. The horizontal line shows the median, the box spans the IQR and whiskers span 1.5 IQR. *n*, number of cells. Statistical test is a one-way ANOVA. The dashed lines mark the inner cell mass (ICM). Note that the ICM of diapaused embryos sometimes polarizes, as reported before[8]. Accompanying stainings can be found in Extended Data Fig. 2. **e**, The accessibility of regulatory elements in ES cells as mapped by ATAC-seq. All accessible regions were determined by peak calling, then clustered into three groups showing high, medium and low accessibility (clusters 1, 2 and 3, respectively). Right panels show the genomic composition of each cluster. Cluster 1 is enriched for promoters; clusters 2 and 3 are enriched for distal regulatory elements. No new peaks were gained during the treatment.

proliferation, which in turn maintains the permissive chromatin environment[11–13]. Transcription itself repels repressive complexes in ES cells[14]. As such, the proliferation rate, chromatin status and pluripotency appear to be interlinked. Yet, uncovered strategies to counteract genomic repression may thus be critical for pluripotency maintenance in dormancy.

DNA methylation is considered a major layer of epigenetic regulation. While CpG islands generally remain unmethylated, other sites in the genome such as repetitive elements, enhancers and non-CpG-rich promoters are subject to DNA methylation, which may alter expression levels. In one of two global DNA demethylation waves, DNA methylation decreases during preimplantation development to its lowest levels in

the blastocyst and sharply rises after implantation[15]. A combination of deposition by DNA methyltransferases (DNMTs) and erasure control DNA methylation levels in cells. Erasure occurs via passive dilution and/or active DNA demethylation by TET enzymes[16,17]. Despite the association of low DNA methylation levels with the naive pluripotent state, TET DNA demethylases are not essential for the specification or maintenance of pluripotent cells, evidenced by the ability of *Tet1/2/3* triple knockout (TKO) mice to generate blastocysts and of *Tet* TKO ES cells to maintain pluripotency in culture[18–21]. TET activity in ES cells comprises catalytic and noncatalytic functions and is associated with pluripotency transcriptional networks and genomic repeats as well

as proximal and distal regulatory elements[22–27]. Due to these highly modular functions, it is essential to investigate occupancy in parallel with catalytic activity to reveal specific mechanisms.

In this Article, we reveal dynamic chromatin adaptations, including DNA demethylation, recruitment of methylation-sensitive transcription factors (TFs) and consolidation of histone modifications, during the cellular transition into dormancy that safeguard cellular identity during dormancy.

## Results

### Genomic rewiring during ES cells' transition to dormancy

To understand the genomic alterations that accompany the transition of pluripotent cells into dormancy, we leveraged our mTORi-induced in vitro diapause model and first profiled gene expression, DNA methylation and chromatin accessibility changes over time (0–144 h) (Fig. 1a and Extended Data Figs. 1 and 2). We have previously shown that mTOR inhibition triggers a multi-level diapause response, with 'immediate' and 'adaptive' steps[6]. Likewise, in vivo-diapaused mouse embryos show dynamic signaling and morphological changes in the first days of the transition, with a complete cessation of proliferation only happening after 5–6 days[8,28]. Based on this knowledge, we rationalized that the full establishment of a robust dormancy program at the genomic level may also require a transition period and, thus, collected time points until 144 h (6 days). Spike-in normalized bulk RNA sequencing (RNA-seq) revealed that the transcriptome was globally repressed already after 24 h, with a further gradual downregulation over time and only a small subset of genes escaping this trend (Fig. 1b, Extended Data Fig. 1a,b and Supplementary Table 1). Starting with ES cells carrying low DNA methylation, we observed a global increase in methylation starting at 72 h, CpG islands being an exception (Fig. 1c and Extended Data Fig. 2a,b). Immunofluorescence (IF) stainings for 5mC in ES cells as well as embryos confirmed this increase (Fig. 1d and Extended Data Fig. 2c,d). DNA methylation levels reverted back to those of normal blastocysts once in vivo-diapaused embryos were reactivated in culture (Extended Data Fig. 2d).

Together with global transcriptional repression and increased DNA methylation, chromatin accessibility at regulatory elements was reduced at 144 h of mTORi treatment (Fig. 1e; assay for transposase-accessible chromatin with sequencing (ATAC-seq) data only inform of regulatory elements, not global accessibility). However, in contrast to transcriptome and DNA methylation changes, chromatin accessibility showed a highly dynamic behavior, with transiently increased accessibility of regulatory elements between 24 h and 72 h of treatment (Fig. 1e; shown are all accessible regions clustered into three groups by accessibility level). Therefore, the cellular transition to dormancy entails a global decrease in transcriptional activity and dynamic chromatin changes.

### TET activity is essential for pluripotency maintenance

DNA methylation increase temporally follows transcriptional repression and probably arises from it. However, it may, in turn, also contribute to repression of genomic activity. To test whether DNA methylation is required for dormancy, we tested the capacity of DNA methyltransferase 3a/3b double knockout (*Dnmt3a/b* DKO) ES cells[29], which lack de novo DNA methylation machinery, to establish the paused pluripotent state (Fig. 2a). Similar to wild-type ES cells, *Dnmt3a/b* DKO ES cells reduced proliferation and decreased global transcription, at the same time maintaining pluripotent morphology (Fig. 2a and Extended Data Fig. 3a,b), suggesting that DNA methylation increase is nonessential for a successful transition into dormancy in this context.

The naive pluripotent state is classically associated with low DNA methylation[15], except in diapaused embryos (Fig. 1d). 5hmC, a DNA demethylation intermediate, increased in paused ES cells along with 5mC (Extended Data Fig. 2c). Therefore, we next asked whether active DNA demethylation is required in dormancy. For this, we tested whether *Tet1/2/3* TKO cells, which we generated from *Tet1/2/3*^flox/flox cells[30], can transition to dormancy under mTORi (Fig. 2b and Extended Data Fig. 3c). Unlike wild-type and *Dnmt3a/b* DKO cells, *Tet* TKO cells failed to maintain ES cell colony morphology under mTORi, with many colonies flattening out over time (Fig. 2b). This phenotype was rescued upon ectopic expression of wild-type, but not catalytic-dead, *Tet1* or *Tet2* in the *Tet* TKO background (Fig. 2c and Extended Data Fig. 3d).

To reproduce these results in an independent cell line, we generated a feeder-independent *Tet1/2* DKO ES cell line using Cas9-assisted deletion of the entire genes (Extended Data Fig. 3e–g). *Tet* DKO ES cells failed to maintain pluripotency under mTORi and differentiated, especially after 72 h, concurrent with the increase in DNA methylation (Fig. 2d and Extended Data Fig. 3h; staining for the activity of the pluripotency marker alkaline phosphatase is shown). Apoptosis levels were only slightly above wild-type cells; thus, mTORi did not induce cell death (Extended Data Fig. 4b,c). Co-depletion of DNMT and DNA demethylase activities rescued the depletion of TET-deficient cells under mTORi (Fig. 2e; refs. 31,32).

To better time-resolve the pluripotency dynamics in wild-type and *Tet1/2* DKO ES cell populations, we measured expression of the pluripotency-associated surface marker SSEA1 via flow cytometry (Fig. 2f and Extended Data Fig. 4a). In wild-type ES cells, SSEA1 expression steadily increased during mTORi treatment, with ~95% of cells positive for SSEA1 at 96 h (Fig. 2f and Extended Data Fig. 4a). *Tet1/2* DKO response copied the wild-type until 48 h but diverged starting at 72 h (Fig. 2f and Extended Data Fig. 4a). RNA-seq likewise revealed the divergent transcriptional state of *Tet1/2* DKO cells starting at 72 h (Fig. 3a,b, Extended Data Fig. 5a,b and Supplementary Table 1). These results show that (1) mTORi treatment shifts wild-type ES cells to a uniform, highly pluripotent state, (2) *Tet1/2* DKO cells are unable to maintain this pluripotent state, and (3) this defect arises concurrent with the increase of DNA methylation and, therefore, (4) TET activity is essential for maintenance of pluripotency during dormancy transition.

### *Tet1/2* DKO ES cells fail to transition to paused pluripotency

To further probe the requirement for TET function in dormancy transition, we next compared DNA methylation, and chromatin accessibility

---

**Fig. 2 | Catalytic activities of TET DNA demethylases are indispensable for the maintenance of pluripotency during dormancy. a**, Proliferation curves and brightfield images of wild-type and *Dnmt3a/b* DKO ES cells (devoid of de novo methyltransferase activity) treated with the mTOR inhibitor INK128 for 120 h. Data are from two biological replicates. Individual data points are shown; lines denote the mean. **b**, Same as in **a** for *Tet1/2/3*^flox/flox versus *Tet1/2/3* TKO iPS cells[30]. *Tet* TKO cells lose pluripotent colony morphology over time under dormancy conditions. **c**, Rescue of *Tet* TKO dormancy defect via overexpression of wild-type, but not catalytic-dead (cd), *Tet1* or *Tet2*. The catalytic-dead mutations can be found in Methods and Extended Data Fig. 3d. Images are representative of two biological replicates. **d**, Alkaline phosphatase staining of an independently generated, feeder-independent *Tet1/2* DKO ES cell line in normal and mTORi conditions. See Extended Data Fig. 3e–g for details of the deletions and accompanying proliferation curves. *Tet1/2* DKO ES cells lose pluripotent colony morphology and marker (alkaline phosphatase) expression during mTORi treatment. The rightmost images are magnifications of the asterisk-marked colonies. Images are representative of two biological replicates. **e**, Rescue of *Tet* TKO dormancy defect in the absence of DNMT activity. Wild-type or *Dnmt* TKO ES cells were treated with the TET inhibitor (TETi) Bobcat339 with or without mTORi. TETi-treated cells are depleted specifically under mTORi treatment in wild-type but not *Dnmt* TKO ES cells. Individual data points are shown; lines denote the mean. **f**, Flow cytometry analysis of SSEA1 expression levels (a pluripotency marker) in wild-type and *Tet1/2* DKO cells in normal and mTORi conditions. Left: overlays of SSEA1 expression at 0 h versus 96 h in wild-type or *Tet1/2* DKO cells. Right: stacked bar plots showing quantification of SSEA1 expression levels at all quantified time points. All flow cytometry plots are shown in Extended Data Fig. 4a. Data from two biological replicates are shown.

patterns of *Tet1/2* DKO and wild-type cells. *Tet1/2* DKO cells were sampled at 0, 24 and 72 h before pluripotency defects became obvious in culture (Fig. 3c). Global DNA methylation was similarly increased in wild-type and *Tet1/2* DKO cells during the course of the mTORi treatment (Fig. 3c and Extended Data Fig. 5c). This suggests that TET activity is probably not required throughout the genome, but rather at specific regulatory elements, consistent with previous findings[29]. Chromatin accessibility at regulatory elements was prematurely reduced already at 72 h in *Tet1/2* DKO cells, compared to 144 h in wild type (Fig. 3d).

Taken together, loss of TET activity compromised the adaptation of pluripotent cells to dormancy conditions. Robust expression of *Tet* genes during dormancy transition further underlines their critical role (Extended Data Fig. 5d,e).

## TETs demethylate DNA at distal regulatory elements

To pinpoint specific genomic sites that require TET activity during the transition into dormancy, we next profiled TET1 and TET2 binding in wild-type cells via CUT&Tag (Fig. 4a,b). Similar to previous studies,

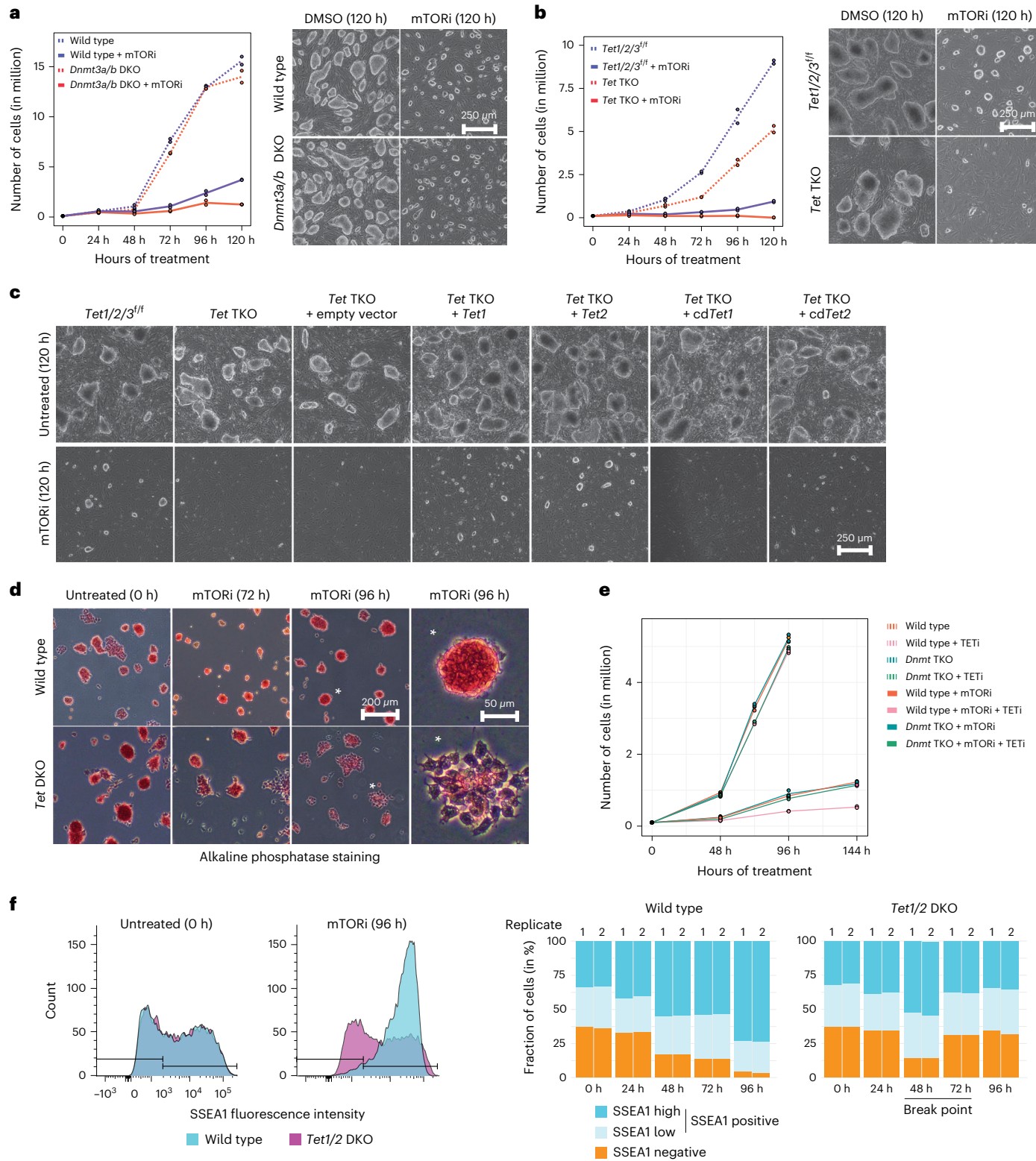

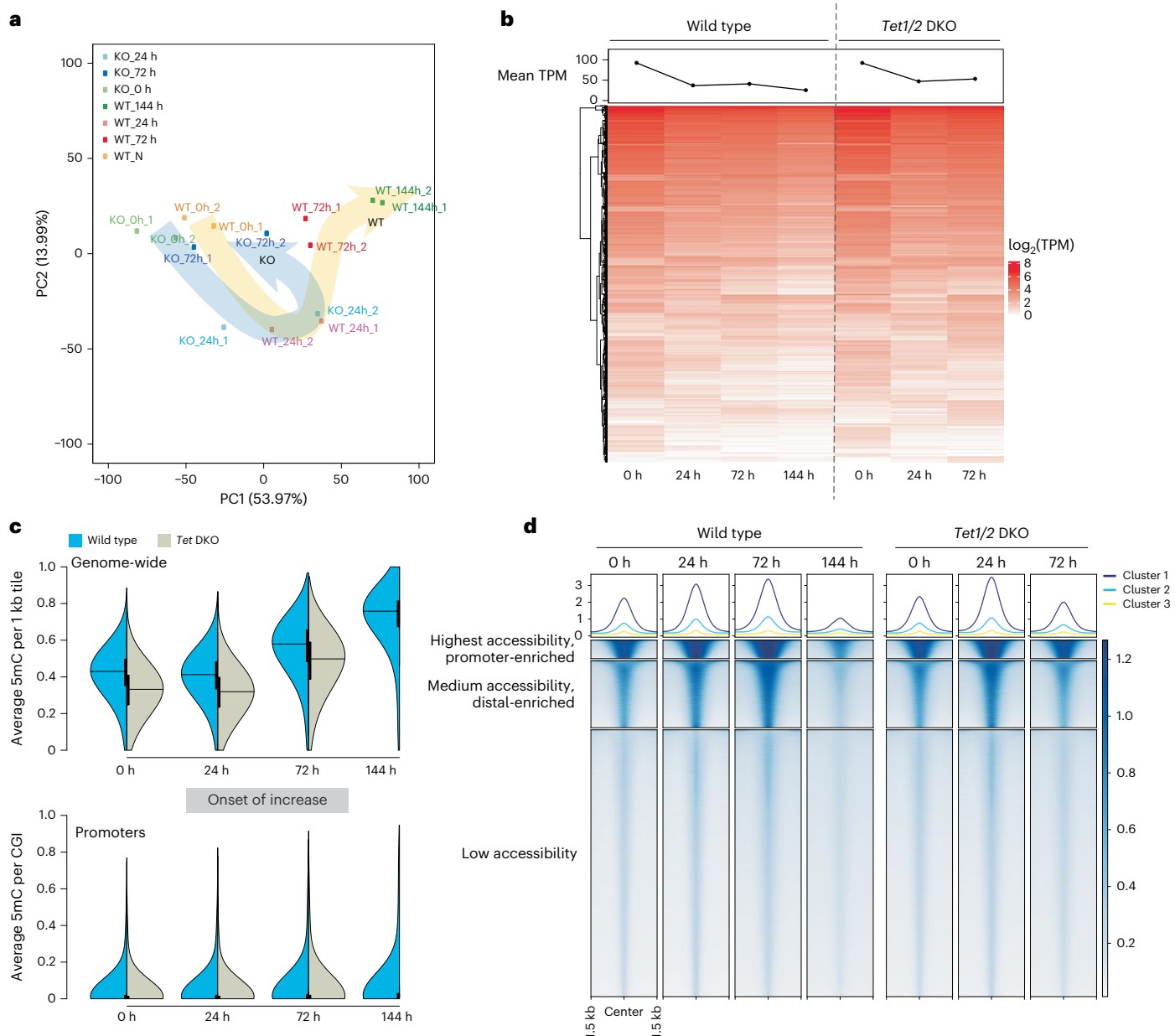

**Fig. 3 | *Tet1/2* DKO ES cells fail to correctly rewire chromatin and install the transcriptional program of paused pluripotency. a**, Principal components (PC) analysis (based on 5,000 most variable genes between 0 h and 144 h wild type) of all RNA-seq samples. *Tet1/2* DKO cells were collected until 72 h before differentiation of colonies. Two replicates were performed for all experiments. Sequencing depth and quality control parameters can be found in Extended Data Fig. 1a. *Tet1/2* DKO cells initiate the dormancy program but fail to fully establish it. **b**, Spike-in normalized bulk RNA-seq heatmap showing expression of all genes over time in wild-type versus *Tet1/2* DKO ES cells treated with mTORi. The line plot on top shows mean TPM at each time point. **c**, DNA methylation levels in wild-type versus *Tet1/2* DKO ES cells as mapped by whole genome bisulfite sequencing. Average DNA methylation levels of the entire genome in 1 kb tiles (top) or CpG islands (CGI) (bottom) are shown. Horizontal lines show the median; box plots within violin plots show interquartile range (IQR), and the whiskers show 1.5 IQR. **d**, The accessibility of regulatory elements in wild-type versus *Tet1/2* DKO ES cells as mapped by ATAC-seq. All accessible regions as determined by peak calling were clustered into three groups showing high, medium and low accessibility (clusters 1, 2 and 3, respectively).

we found that TETs bind thousands of regions throughout the genome (Fig. 4a). However, the vast majority of them show no further increase in DNA methylation in their absence. To identify sites where the DNA demethylation activity of TETs is relevant, we filtered the TET-bound targets to those that (1) are protected from methylation increase in wild-type cells and (2) show at least a 10% increase in methylation in *Tet1/2* DKO cells compared to wild type at 72 h (Fig. 4a). We identified 5,164 such 'TET-dormancy targets', which TET1 and/or TET2 keep lowly methylated despite the global increase in methylation (Fig. 4b–e and Extended Data Fig. 6a). Among these, 1,223 are bound by both TET1 and

TET2, 3,646 are bound only by TET1 and 295 are bound only by TET2 as detected by our methods (Fig. 4b). Among 'TET-dormancy targets' we find enriched ES cell enhancers (10% and 13% of targets are active and primed enhancers, respectively) and repetitive elements, particularly of the L1Md family of LINE1 repeats (33% of targets) (Fig. 4b). The 'Other' cluster contains a variety of repetitive elements other than L1Md family LINE1 repeats.

TET-dormancy targets, an example of which is shown in Fig. 4e, are demethylated in normal proliferative ES cells, in both wild-type and *Tet1/2* DKO backgrounds. Therefore, TET activity is dispensable

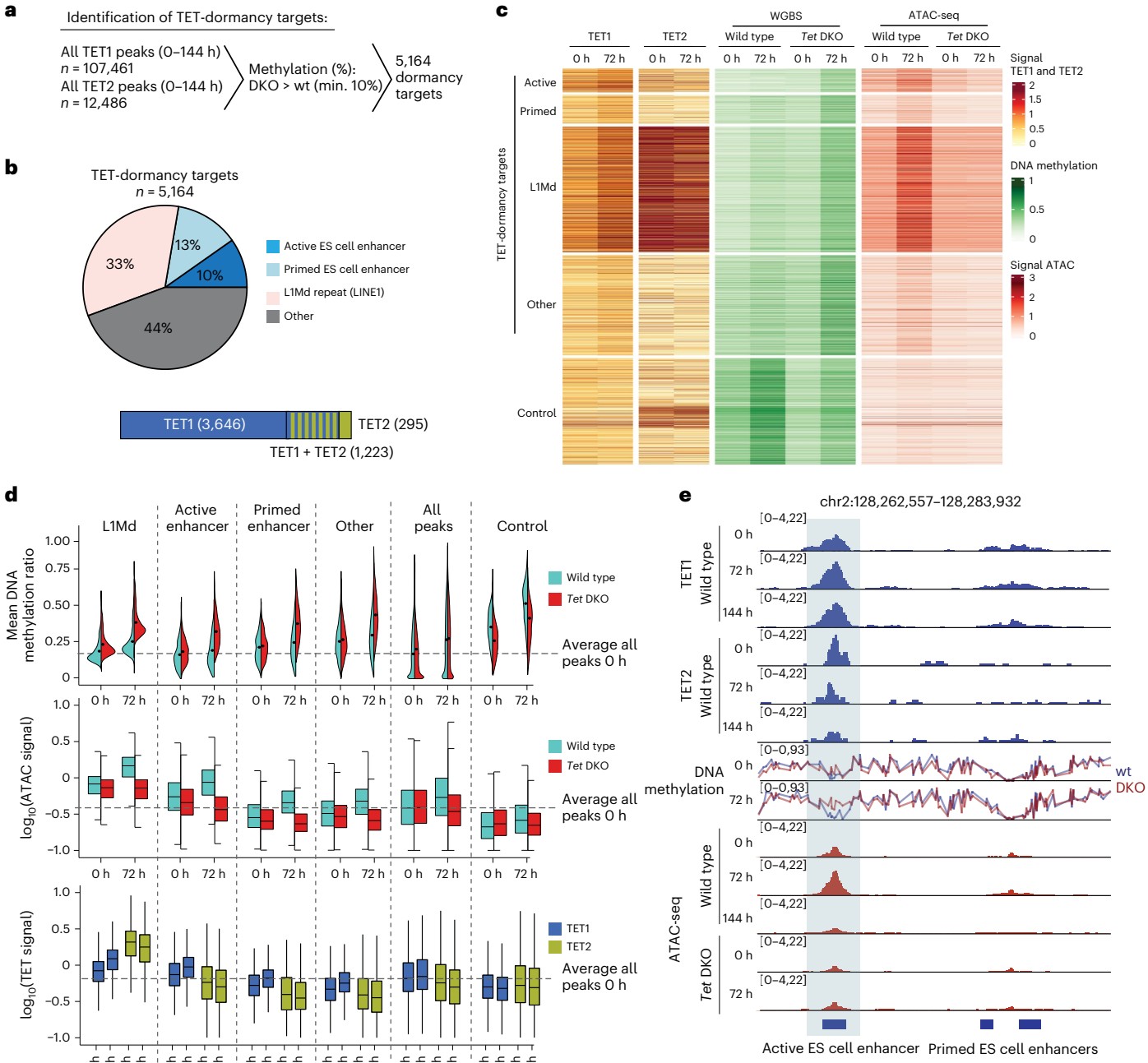

**Fig. 4 | TETs counteract DNA methylation at ES cell enhancers and young LINE1 elements. a**, Identification of TET-dormancy targets. Sites that are bound by TET1 and/or TET2 in the wild type (wt) and that are kept demethylated by TET activity (that is, methylation is increased only in the *Tet1/2* DKO at a minimum (min.) of 10%) are specified as targets. **b**, Pie chart showing the distribution of different genomic features within TET-dormancy targets. The bar plot shows TET1- and/or TET2-bound targets as determined by CUT&Tag. The 'Other' cluster contains a variety of repetitive elements other than L1Md family LINE1 repeats, with no specific enrichment. **c**, A heatmap showing mean levels of TET occupancy, DNA methylation and chromatin accessibility at TET-dormancy

targets in wild-type versus *Tet1/2* DKO ES cells over time during mTORi treatment. Control: 2,000 random sites with increased DNA methylation in wild-type cells at 72 h compared to 0 h. Accompanying heatmaps showing signal distirubution can be found in Extended Data Fig. 6a. **d**, Quantification of data shown in **c** versus all TET peaks. The dashed lines show the median signal at all TET peaks in wild-type ES cells at 0 h. The horizontal lines show the median; the box plot within violin plots shows interquartile range (IQR), and the whiskers show 1.5 IQR. **e**, Genome browser view of an example TET-dormancy target active enhancer and a neighboring nontarget primed enhancer.

at these sites under normal ES cell culture conditions, and probably in normal blastocysts. However, a specific dependency on TET activity arises in dormancy. Taken together, we conclude that TET catalytic activity is required to protect a subset of ES cell enhancers and L1Md repeats from increase in DNA methylation, which may be detrimental to their function during or after dormancy.

## TF accumulation at TET-dormancy targets
Why is TET activity required at these specific targets during the transition to dormancy? We hypothesized that DNA demethylation by TET enzymes could alter the binding of methylation-sensitive TFs at dormancy targets, thereby potentially affecting the activity of the regulatory elements. To address this possibility, we first performed TF

motif enrichment analysis of the underlying DNA sequence (Fig. 5a). TF motif enrichment analysis revealed that a notable proportion (~30%) of TET-dormancy targets have binding sites for methylation-sensitive TFs such as YY1, TFE3, ZFP57, and KLF4 (refs. [31–34]). In contrast, core pluripotency factors, some of which have been shown to interact with TETs (for example, NANOG[35]), are not notably enriched above background at TET-dormancy targets.

To further probe potential TF binding at TET-dormancy targets, we examined TF 'footprints' in ATAC-seq data[36]. Candidate TFs (from Fig. 5a), but not canonical pluripotency-associated TFs, showed increased footprints in wild-type cells during dormancy transition, but not as much in *Tet1/2* DKO cells (Fig. 5b). In contrast to candidate TFs, pluripotency TFs showed reduced footprints, particularly at 144 h, in wild-type ES cells, hinting at the shutdown of the transcriptional network of pluripotency. Therefore, pluripotency is maintained and even enhanced (Fig. 2f) in the paused state, even though the transcriptome is largely repressed and canonical pluripotency TFs may be disengaged from their chromatin targets. These observations bring to surface the unique regulation of pluripotency in dormancy.

To test whether the above computational predictions represent actual changes in TF binding, we chose TFE3 from among the expressed candidate TFs and profiled its binding at (1) TET-dormancy targets, (2) control regions that are bound by TETs but do not show increase in methylation and (3) canonical TFE3 targets that are bound by TFE3 but are not among TET-dormancy targets (Fig. 5c–e and Extended Data Fig. 6b,c). TFE3 is a nutrient-sensitive TF that has been shown to translocate into the nucleus in response to inhibition of the PI3K/mTOR pathway, making it a highly interesting candidate TF in the context of mTORi-induced dormancy[37–40]. Additionally, TFE3 has been recently predicted to be a methylation-sensitive pluripotency TF via a single-molecule multi-omics approach[41]. Genome-wide profiling revealed TFE3 accumulation at TET-dormancy targets in wild-type cells at 72 h and further at 144 h of mTORi treatment (Fig. 5c–e). Neither control regions nor canonical TFE3 targets showed increased TFE3 binding, suggesting that presence of TETs alone does not suffice to increasingly recruit this TF. TET-dormancy targets showed less TFE3 binding in *Tet1/2* DKO cells already at 0 h and failed to accumulate it over time (Fig. 5c,d and Extended Data Fig. 6c). Notably, TFE3 levels remained high in wild-type cells at 144 h of treatment, despite lower chromatin accessibility (Fig. 5c–e). Thus, the globally less active genome of dormant cells contains sites that remain bound by TETs and TFE3 at 144 h.

To corroborate TET–TF interactions strongly suggested by our analysis and test whether TETs and TFs interact biochemically, we performed native (no crosslinking) immunoprecipitation (IP)–mass spectrometry (MS) using an ES cell line that carries a Flag tag at the endogenous *Tet1* locus[26] (Extended Data Fig. 6d). Untagged wild-type cells were used as control. Via this approach, we detected 553 proteins that were enriched only in the *Tet1*-Flag cell line and not in wild-type cells (Fig. 5f and Supplementary Table 2). TFE3 and YY1, along with the known TET1 interactors such as SIN3A, were detected in both normal

and mTORi ES cells (Fig. 5f). These data support a model where TETs and methylation-sensitive TFs may co-occupy genomic targets in dormant cells.

Finally, to test whether TFE3 itself is functionally relevant for the maintenance of pluripotency in dormancy, we generated an inducible *Tfe3*-knockdown ES cell line to selectively deplete TFE3 during dormancy transition (Fig. 5g). TFE3 was fully depleted upon doxycycline treatment (Fig. 5h). Depletion of TFE3 in normal ES cell culture did not change the colony morphology or activity levels of the pluripotency marker alkaline phosphatase yet led to differentiation of ES cell colonies under dormancy conditions (Fig. 5i). Overall, these data suggest that TETs and TFE3 interact biochemically and genetically to safeguard pluripotency as cells transition into dormancy.

## L1Md expression and TET binding in dormancy

TFE3 accumulated above-average levels particularly at active L1Md repeats and at active enhancers. We then asked whether increased TET and TFE3 binding leads to increased transcriptional output of these targets. We first examined expression levels of all L1Md repeats by mapping RNA-seq data from wild-type and *Tet1/2* DKO cells to the consensus sequence of each repeat (Fig. 6a). Expression of L1Md repeats was higher in wild-type cells compared to *Tet1/2* DKO at 0 h. Notably, L1Md_Tf, L1Md_Gf and L1Md_A repeats, which are the specific elements bound by TETs, were transiently upregulated at 24 h of treatment in wild-type cells, while most other L1Md elements as well as other LINEs and LTRs were not (Fig. 6a). The upregulated repeats are evolutionarily younger[42] elements (Fig. 6b) that have intact 5′ ends that allow transcription[42–44]. Their transient upregulation depends on TET activity, as revealed by their inertness in *Tet1/2* DKO cells. In striking contrast, most other L1Md elements were upregulated in *Tet1/2* DKO cells at 72 h of treatment, signaling overall transcriptional deregulation (Fig. 6a).

TET–RNA interactions have been documented to mediate cellular transitions[45]. Therefore, we next asked whether TETs bind the upregulated L1Md RNAs in cells transitioning into dormancy. To capture RNAs bound to TET1/2 proteins, we knocked in biotin-receptor tags into endogenous *Tet1/2* loci and performed FLASH (fast ligation of RNA after some sort of affinity purification for high-throughput sequencing)[46] (Fig. 6c and Extended Data Fig. 7a,b). TETs and interacting RNAs were isolated after crosslinking via streptavidin binding and stringent washes (Fig. 6c). Wild-type cells were used as control. FLASH revealed increased binding of L1Md repeat RNAs to TET1 and TET2 at 72 h or 144 h, but not at 24 h (Fig. 6d,e and Extended Data Fig. 7c). Since these repeats are highly transcribed at 24 h, their binding to TETs is probably not co-transcriptional. In contrast, other repeats that are transcribed at 72 h such as IAPLTR1a_Mm did not show TET binding. Overall, the most abundant repeat RNAs that interact with TETs belonged to L1Md_A, L1Md_T, and L1Md_F elements (Fig. 6e). LINE–TET interactions may promote or stabilize TET occupancy and regulate chromatin accessibility at 72 h.

**Fig. 5 | TET activity at dormancy targets mediates TF binding. a**, TF enrichment analysis at TET-dormancy targets. The presented motifs are all significant with *P* value <0.0001. **b**, TF footprinting[36] analysis of predicted TET-activity-coupled TFs versus classical pluripotency TFs. ATAC-seq signal from wild-type and *Tet1/2* DKO cells was used. TFE3, YY1 and ZFP57 footprints are elevated and remain high at 144 h in wild-type cells compared to *Tet1/2* DKO cells. In contrast, footprints of pluripotency-associated TFs are reduced. Significance (*P* values) of the binding activity of TFs was derived with the BINDetect function of the TOBIAS package. **c**, Levels of TFE3 binding at TET-dormancy targets versus controls, mapped by CUT&Tag. TFE3 occupancy increases over time specifically at sites that are kept demethylated by TETs, and particularly at L1Md repeats and active enhancers. Accompanying quantifications are in **d** and Extended Data Fig. 6b. **d**, Quantification of data shown in **c** versus canonical TFE3 targets (as identified by peak calling at *t* = 0 h). The dashed lines show the median TFE3

signal in wild-type (WT) ES cells at 0 h. Statistical test is a one-way ANOVA with Tukey's multiple comparison test. Horizontal lines denote the median, and lower and upper hinges denote the first and third quartiles. The whiskers denote 1.5 times the interquartile range. **e**, Genome browser view of an example TET-dormancy L1Md repeat showing TET and TFE3 occupancy, DNA methylation and genome accessibility. **f**, Proteins co-precipitated with TET1, as identified by IP–MS. Label-free quantification (LFQ) values are plotted. **g**, The experimental outline of inducible TFE3 knockdown (*Tfe3* iKD) and mTORi treatment. **h**, IF images showing efficient knockdown of TFE3 expression after 24 h of doxycycline (dox) treatment. The images are representative of two biological replicates. **i**, Alkaline phosphatase staining of control or *Tfe3* iKD ES cells with or without mTORi treatment. *Tfe3* iKD ES cells lose pluripotent colony morphology and marker expression after 72 h of mTORi culture. The images are representative of two biological replicates.

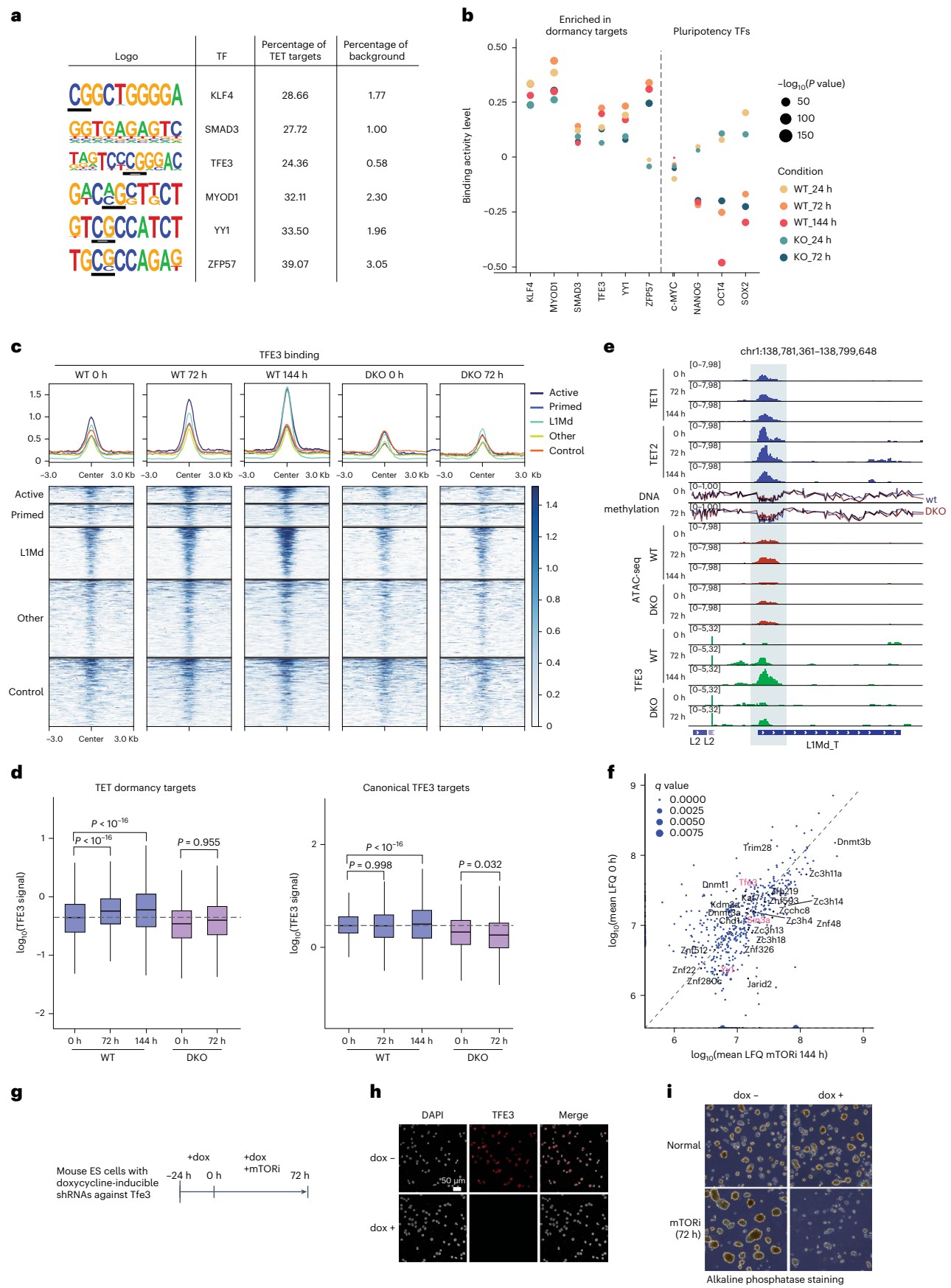

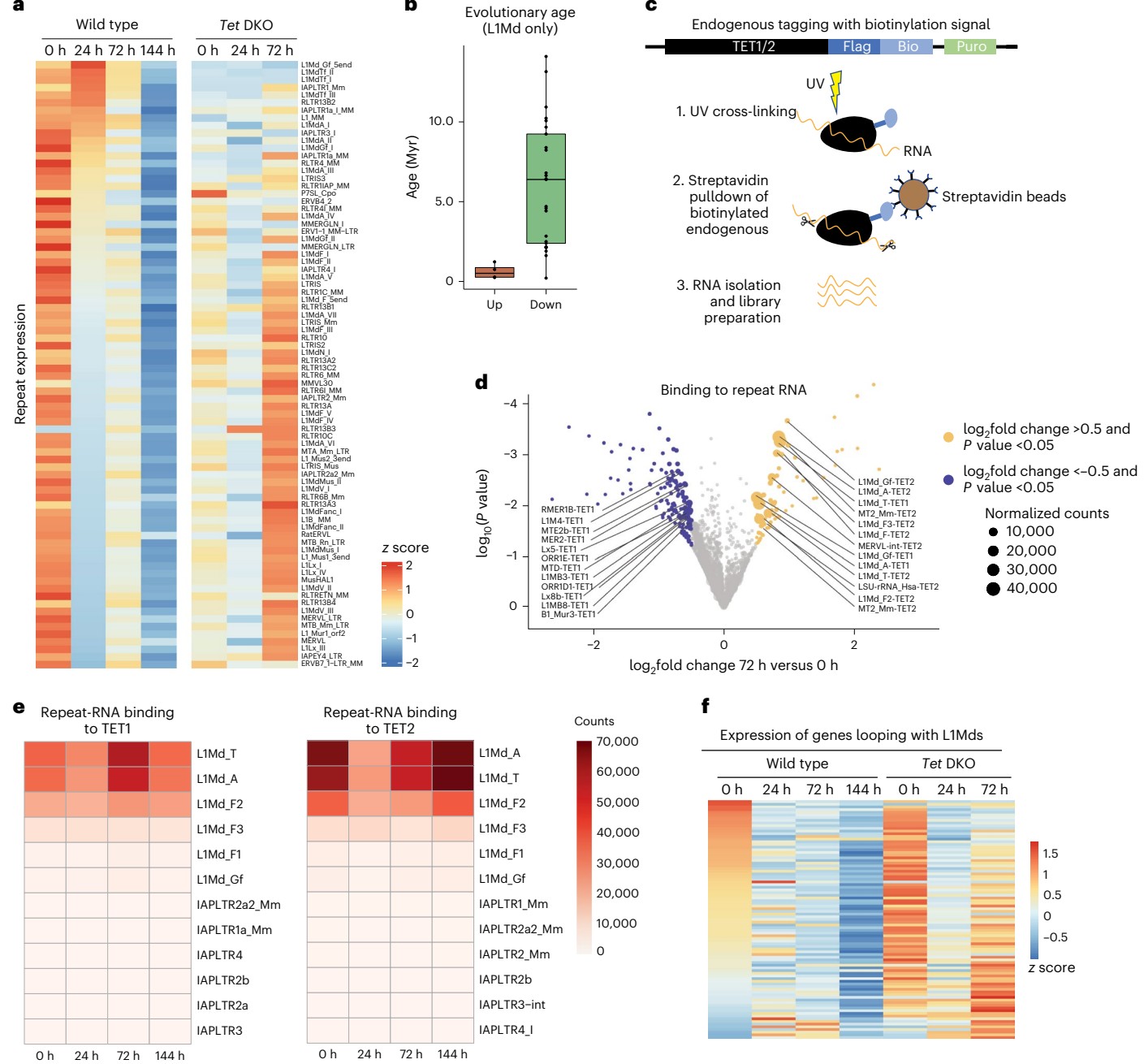

**Fig. 6 | Expression and TET binding of young LINE1 elements in ES cells during the transition into dormancy. a**, The expression levels of repetitive elements in wild-type and *Tet1/2* DKO ES cells at indicated time points of mTORi treatment. Reads were mapped to the consensus sequence of each repeat retrieved from RepBase[69]. L1Md repeats are transiently upregulated at 24 h of mTORi treatment. **b**, The evolutionary age of LINE1 repeats that are upregulated versus downregulated at 24 h of mTORi in wild-type cells. The horizontal lines denote the median, and lower and upper hinges denote the first and third quartiles. The whiskers denote 1.5 times the interquartile range. The dots show individual values. **c**, Schematics of the FLASH[46] experiment to map TET-bound RNAs.

The protocol allows stringent washes due to streptavidin-mediated capture. **d**, A volcano plot showing RNAs that are differentially bound to TET1/2 at 72 h compared to 0 h of mTORi versus 0 h. Both TET1 and TET2 bind L1Md repeats. Differential binding at 24 and 144 h is shown in Extended Data Fig. 7c. *P* values and log$_2$ fold changes were derived using DEseq2. **e**, Binding of L1Md and IAPLTR repeat RNAs to TET1/2 at all time points. Note that IAPLTRs are also transiently upregulated (shown in **a**) but do not bind TETs. **f**, The expression levels of genes in putative contact with L1Md promoters. Contact was determined by analysis of published HiC datasets[49] (Methods) with a contact probability >15. TET activity at L1Md elements is uncoupled from transcription during dormancy.

LINE1 repeats have been proposed to function as enhancers in ES cells[47,48]. To probe whether TET binding at LINE1 repeats and de facto ES cell enhancers induce transcription, we set out to identify genes that may be regulated by these elements. By using HiC contact maps and H3K27 acetylation tracks[49], we isolated gene promoters putatively looping with TET-dormancy targets[49] (see Methods for details). These putative target genes were mostly downregulated at 72/144 h (Fig. 6f and Extended

Data Fig. 7d). As such, TET/TF activity at dormancy targets appears to be largely uncoupled from gene expression and may instead poise them for use at a later time point, namely after release of cells from dormancy.

## Chromatin dynamics at TET-dormancy targets
To investigate the possibility of TET/TF-mediated chromatin poising, we next investigated chromatin dynamics at TET-dormancy targets over

time during entry and exit from dormancy ('pause' versus 'release'). The transcription-regulatory histone modifications H3K4me1, H3K27ac and H3K4me3 were profiled (Fig. 7 and Extended Data Figs. 8a,b and 9a–c). While control regions showed stable levels of these marks over time, TET-dormancy targets underwent dynamic changes within the first 96 h of dormancy entry and first 48–72 h of release (Fig. 7a and Extended Data Fig. 8a). H3K27ac and H3K4me1 levels increased at TET-target active ES cell enhancers during pausing in wild-type cells, while *Tet1/2* DKO cells showed a muted response (Fig. 7a,b). The TET-target active enhancers retained these marks, at original levels or higher, throughout the duration of pausing despite global transcriptional repression (Fig. 7a,b and Extended Data Figs. 8b and 9). In contrast, H3K27ac reduced at other active ES cell enhancers over time (Fig. 7c,d). TET targets responded dynamically upon release in wild-type, but not *Tet1/2* DKO, cells (Fig. 7d). Paused-then-released *Tet1/2* DKO cells showed earlier compromise in SSEA1 expression compared to *Tet1/2* DKO cells in pause (starting at 48 h versus 72 h), suggesting that TET activity may also contribute to pluripotency maintenance at reactivation (Fig. 7e).

### Modulating TET activity affects embryo pausing

Finally, we investigated the functional requirement for TETs directly in embryos (Fig. 8). For this, we combined genetic and pharmacological loss- and gain-of-function approaches to modulate TET activity and scored embryo survival duration and efficiency in the dormant state. First, we generated *Tet1/2/3* TKO or *Tet1/2* DKO embryos by electroporating zygotes with Cas9–guide RNA (gRNA) mixes (Fig. 8a and Extended Data Fig. 10a). Knockout (KO) validation was done by (1) transferring the TKO embryos to surrogate females and embryo phenotyping at E8.5 and (2) quantitative reverse transcription polymerase chain reaction (RT–qPCR) (Extended Data Fig. 10b,c). This approach yielded at least heterozygous loss of all *Tet* genes in all tested embryos (Extended Data Fig. 10b). The resulting embryos showed delayed or arrested development at post-implantation stages as shown earlier[18,19] (Extended Data Fig. 10c). Blastocyst formation was not compromised.

*Tet* DKO embryos showed limited capacity to establish dormancy in both in vitro and in vivo diapause models (Fig. 8b,c). *Tet* TKO recapitulated this finding, albeit less significantly, which may be due to less efficient Cas9-assisted targeting in this setup (Extended Data Fig. 10d). A similar but more pronounced effect was seen when embryos were treated with the TET inhibitor Bobcat339, which blocks its capacity to convert 5mC to 5hmC[50,51] (Fig. 8d and Extended Data Fig. 10e). Survival was further compromised when embryos were pretreated with Bobcat339 for 12 h before the start of mTORi treatment at the blastocyst stage, underlining the importance of TET activity in capacitating cellular transitions to come (Fig. 8e). Finally, supplementing embryos with the TET cofactors α-ketoglutarate or vitamin C[52] (Fig. 8f) increased embryo survival rates, particularly within the first 4–5 days of mTORi treatment, to over 80% (Fig. 8f). These results show that modifications to embryo culture media based on discovered mechanisms of dormancy can improve the efficiency of in vitro diapause. Overall, we describe here a mechanism by which TET DNA demethylases, together

with TFs, mediate chromatin adaptations that ensure maintenance of pluripotency throughout dormancy and may restart pluripotency programs at reactivation.

## Discussion

Embryonic diapause has remained an enigma since its discovery in 1854. Recent progress[3–6,8,28,53–57] started to illuminate the regulation of dormancy at the cellular level, yet we are missing answers to at least two critical questions: (1) Is low anabolic activity sufficient to induce diapause at the cellular level? (2) If not, which other mechanisms ensure faithful propagation of this dormant state? We find here that TET DNA demethylase activity is required to counteract the increase in DNA methylation at a set of regulatory elements in dormancy. Importantly, although *Tet1/2* DKO ES cells initially show signs of a successful transition into dormancy including a transcriptional profile similar to wild-type ES cells, they fail to establish the stable paused pluripotent state. The onset of this failure appears to be the increase in DNA methylation. In the absence of TET activity, ES cell regulatory elements lack dynamic chromatin adaptations including consolidated enhancer marks and recruitment of methylation-sensitive TFs, resulting in loss of pluripotency (Fig. 8g). These results support a model of diapause that requires more than just decreased anabolic activity.

Increase in DNA methylation normally occurs after implantation of the blastocysts into the uterus and is thought to be incompatible with naive pluripotency. We observed in both in vitro- and in vivo-diapaused embryos increased DNA methylation compared to normal blastocysts. Thus, diapause features a mismatch between the methylation status and the developmental stage of the embryo. Interestingly, methylation decreases after reactivation of the diapaused embryo, suggesting that high DNA methylation may indeed be incompatible with peri-implantation events. Based on our results, TET activity is probably necessary during diapause and in the early reactivation phase before DNA methylation reaches its normal levels. In this context, it would be informative to block the decrease of DNA methylation during reactivation to test whether a methylated blastocyst can implant and resume development.

Why is TET activity required specifically at the identified dormancy targets? Through TF motif and footprinting analyses and experimental validations, we show the accumulation of TFE3 at sites that are demethylated by TETs, specifically in dormancy conditions. Since this binding is diminished in *Tet1/2* DKO ES cells, we conclude that the DNA demethylase activity of TETs is required for the binding of these TFs. In most cases, binding and activity of TETs as well as TFs are linked to accessible chromatin and active transcription. Unlike this conventional function, increased TET activity, TF accumulation and the subsequent enrichment of histone marks do not appear to drive gene expression in the paused state. Yet, TET and TF binding are retained even at 144 h. Due to the transcriptional inertness of these bound factors, we hypothesize that the TET/TF accumulation at dormancy targets may 'bookmark' these sites either to protect critical pluripotency regulatory elements from permanent silencing and/or to poise them for reuse at the onset of

**Fig. 7 | Chromatin dynamics at TET-dormancy targets during dormancy entry and exit. a**, Levels of the indicated histone marks at TET-target active enhancers and L1Mds as well as control regions in wild-type and *Tet1/2* DKO cells over a 144 h time course of mTORi-mediated pausing and release. *Tet1/2* DKO cells were paused for 72 h and then released to avoid loss of pluripotent colonies. The lines show mean values, and the shading shows the confidence interval. The dashed lines denote levels of each mark at 0 h in the color-corresponding genetic background. Extended Data Figs. 8 and 9 contain extended overviews of other TET-dormancy targets and pausing durations up to 15 days. **b**, Genome browser view of chromatin dynamics at the same active ES cell enhancer shown in Fig. 4e. The enhancer fails to accumulate H3K4me1 and H3K27ac as well as TFE3 in *Tet1/2* DKO cells during pausing and at release. **c**, Levels of shown enhancer marks in all active ES cell enhancers excluding TET-dormancy targets. H3K27ac levels

decline below 0 h after 96 h and only reach original levels at 120 h of release. The lines show mean values, and the shading shows the confidence interval. **d**, Quantifications of the shown enhancer marks in all versus TET-target active enhancers. The dashed lines denote levels of each mark at 0 h in the color-corresponding genetic background. TET targets acquire acetylation earlier than all enhancers in released wild-type (wt) cells and show larger deficit in *Tet1/2* DKO. The vertical lines of box plots denote the median, and lower and upper hinges the first and third quartiles, respectively. Whiskers extend no further than 1.5 times the interquartile range from the lower and upper hinge, respectively. **e**, Flow cytometry analysis of the pluripotency marker SSEA1 during pausing and after release in wild-type and *Tet1/2* DKO cells. At 48 h of pausing, *Tet1/2* DKO cells appear similar to wild type in SSEA1 expression pattern, but already show defective reactivation.

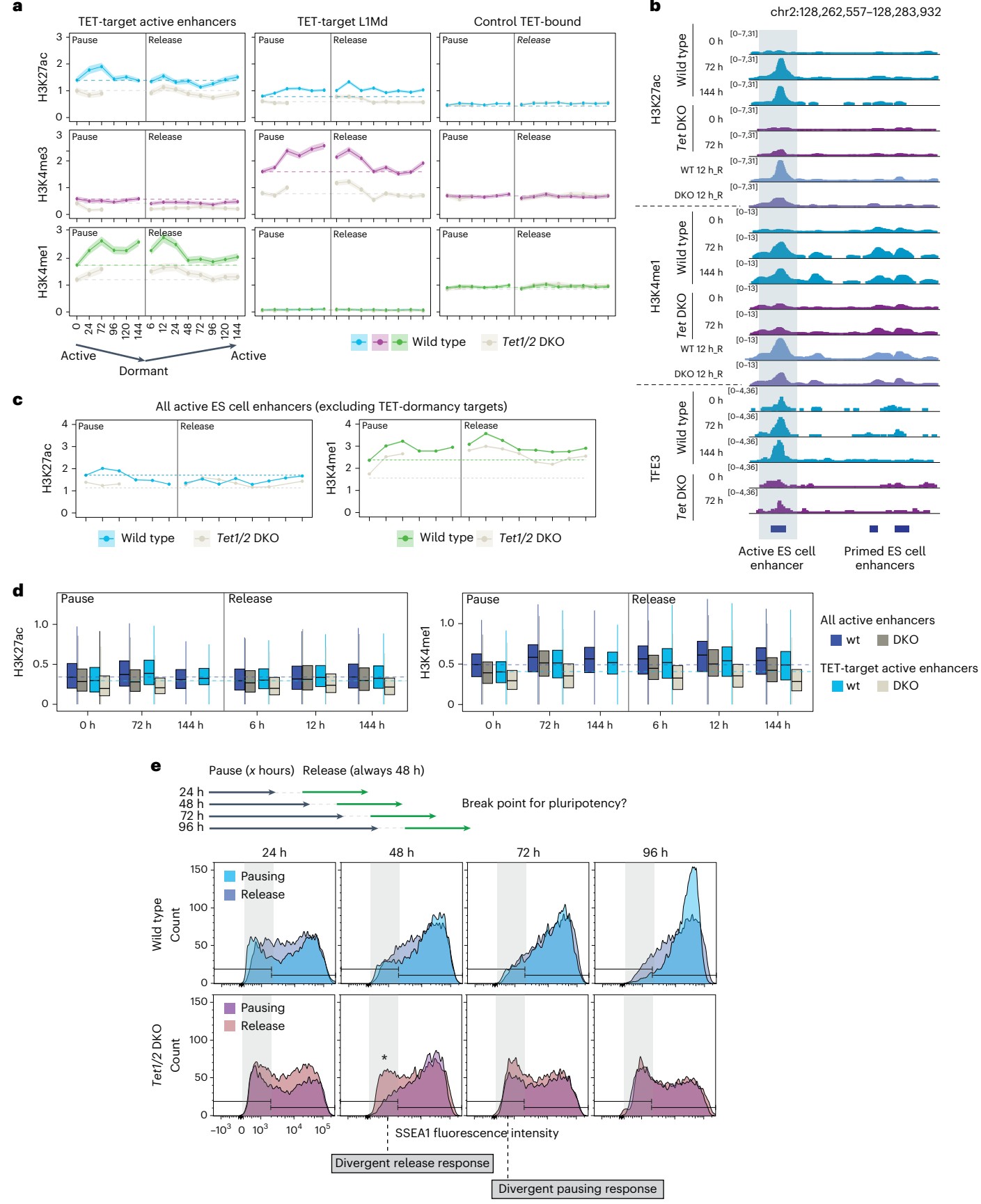

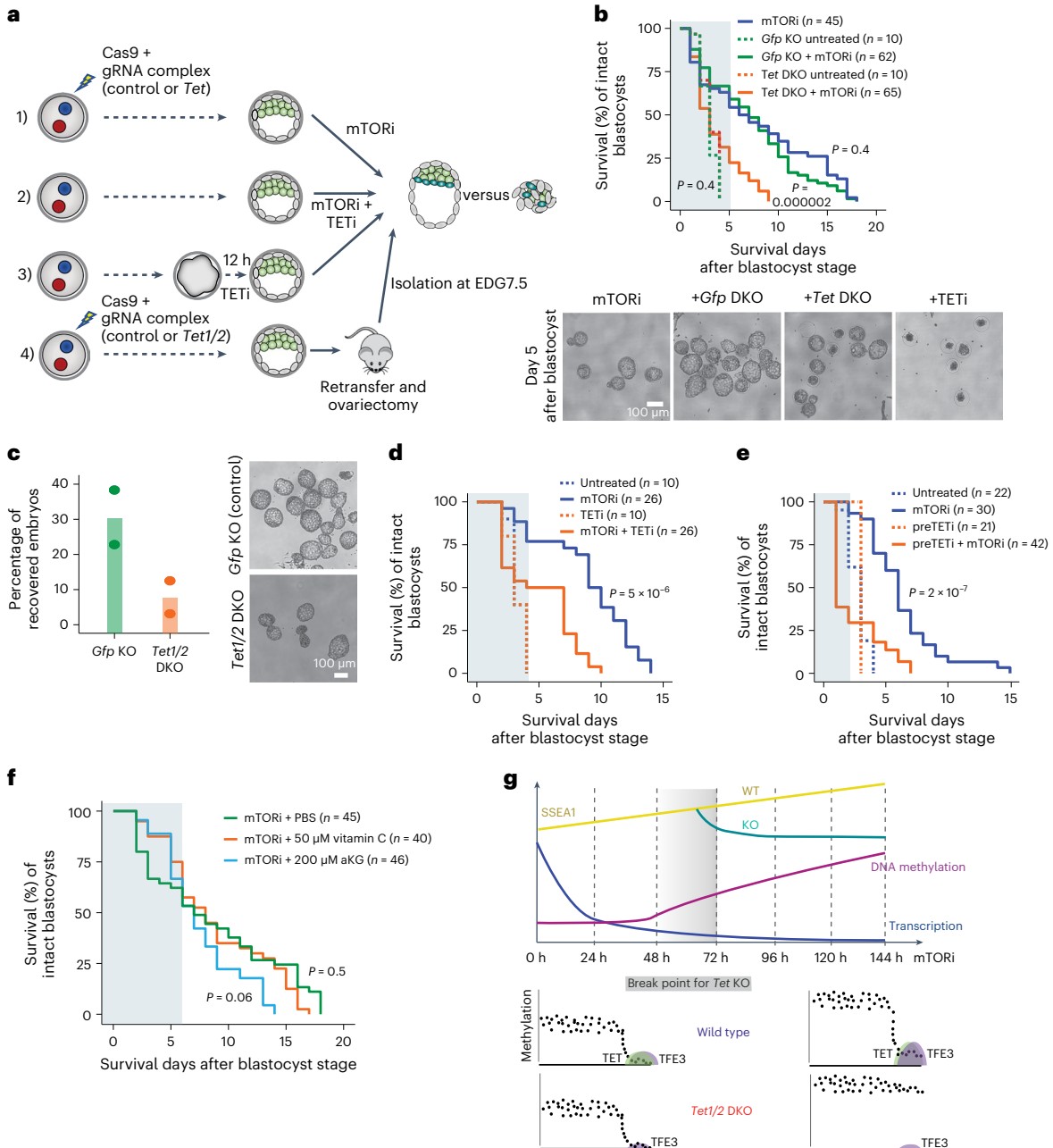

**Fig. 8 | Loss- and gain-of-function perturbations underline the requirement for TET activity in diapause. a**, Workflow of genetic and pharmacological TET loss-of-function experiments. **b**, Top: survival curves of *Tet* DKO and control blastocysts in culture. *n*, number of embryos used in each experiment. Statistical test is log-rank test (R package survdiff) comparing each condition to mTORi-only pausing. The time window in which wild-type and *Tet* loss-of-function embryos show the most divergent response is highlighted. Bottom: brightfield images of representative embryos captured on indicated days of pausing. The same criteria are applied in **d**–**f**. **c**, Left: in vivo diapause efficiency of retransferred control or *Tet1/2* DKO blastocysts. TET deficiency significantly

reduces the recovery rate after in vivo diapause. Right: representative brightfield images. **d**, Survival curves of control, TETi- and TETi + mTORi-treated blastocysts in culture. *n*, number of embryos used in each experiment. **e**, The same as in **d** but with pretreatment of embryos with TETi for 12 h before blastocyst stage. **f**, Survival curves of mTORi-treated blastocysts supplemented with TET cofactors. **g**, Model summarizing the global changes in transcription, DNA methylation and pluripotency status during the transition into dormancy. Bottom panels illustrate the locus-specific regulation at TET-dormancy targets and include DNA methylation levels, TET and TFE3 binding in wild-type and *Tet1/2* DKO cells.

reactivation. Genomic bookmarking is employed to propagate transcriptional programs through mitosis[58,59]; however, bookmarking for longer periods has not been documented. It will also be of high interest to test whether a similar mechanism may be used by adult tissue stem cells to support dormancy-reactivation cycles that enable tissue regeneration.

TET-dependent DNA demethylation of enhancers has been shown in the recent years to mediate several cellular transitions,

from hormone response[60] to cellular reprogramming[61] and differentiation[23,62,63]. TET loss of function led to enhancer methylation and compromised cellular adaptation in several of these cases; therefore, TETs are evidently critical for timed deployment of enhancers at cell state transitions. Prevalent among the regions regulated by TETs in this fashion are ES cell enhancers and evolutionarily young LINE1 elements. Like enhancers, young copies of LINE1 repeats were shown to be bound

by TETs, albeit with no effect on the transcription of the repeat itself[22]. It has been proposed that 5′ ends of young LINE1 repeats function as active enhancers in ES cells[47,48]. By analyzing HiC contact maps[49], we did detect potential looping between TET-target LINEs and gene promoters, but these interactions did not induce the expression of target genes. Young LINE1 elements play a role in decondensing chromatin in early mammalian development[64], and thus, they may play a role in the chromatin adaptations described here.

The necessity for TET activity at enhancers was mechanistically illuminated in the recent years by two groups that showed a methylation–demethylation arms race between TETs and DNMTs[14,27]. DNA methylation levels of enhancers, as determined by the outcome of the arms race, define the TF profile of the enhancer by modulating the engagement of methylation-sensitive TFs. Indeed, the main cause of repression by 5mC at distal elements was recently revealed to be altered TF binding, as opposed to binding of methyl-recognizing repressive proteins such as methyl-CpG-binding proteins[65]. Our results support this model by showing diminished binding of TFE3 at TET-targets in *Tet1/2* DKO cells. A more general multi-omics analysis also identified sensitivity to DNA methylation as a major determinant of binding strength for several TFs[41]. In support of our findings, TFE3 was identified as a methylation-sensitive pluripotency-associated TF with increased binding to unmethylated sequences[41]. Our TF motif enrichment and footprinting analysis suggests that YY1 and ZFP57, two known regulators of LINE1 elements that are also methylation sensitive[24,32,33], may behave similarly to TFE3 in marking LINE1 repeats in dormancy. Thus, even though we focus only on TFE3, our findings are likely to be applicable to other TFs. Targeting dormant cells is one of the main frontiers in improving the prognosis of patients battling cancer. It has been recently shown that cancers dormancy resembles the signature of diapause[66,67]. Our data suggest that the distinct TF footprints of dormant cells can potentially be used to detect, distinguish and even target dormant cells in disease.

Our findings of the increased efficiency of in vitro diapause via supplementation of TET cofactors support the notion that ES cells can be used as a discovery tool to improve embryo cultures. We have recently used an analogous approach to identify metabolic constraints on in vitro-paused embryos[6]. These two independent cases of improving in vitro diapause with metabolite supplementations encourage to use ES cells/induced pluripotent stem (iPS) cells of other species to identify conditions that may support embryo culture and pausing.

In summary, our findings highlight the two-faceted nature of dormancy: (1) elimination of anabolic activity to conserve energy, which is an expected outcome of mTOR inhibition, and (2) active rewiring of chromatin to tolerate the genome silencing that arises as a result of step 1. Our results showing the requirement of active clearance mechanisms are a testament to the dynamic nature of the dormant state of diapause.

## Online content

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

## Methods

### Animal experimentation

Animal experiments were performed according to local animal welfare laws and approved by authorities (Landesamt für Gesundheit und Soziales), covered by LaGeSo licenses ZH120, G0284/18, G021/19 and G0243/18-SGr1_G. Mice (7–12 weeks old) were housed with enrichment material in ventilated cages (humidity 45–65%, temperature 20–24 °C) on a 12 h light/dark cycle and fed ad libitum.

### Cell lines and culture conditions

Wild-type, *Tet1/2* DKO and Tet1^Flag/Tet2^V5/Tet3^HA ES cells[26] (all E14 background) were cultured without feeders on gelatin-coated plates (0.1%, Sigma-Aldrich G1393) at 37 °C in a humidified 5% $CO_2$ incubator with medium containing Dulbecco's modified Eagle medium high glucose with GlutaMAX (Thermo, 31966047), 15% fetal bovine serum (Thermo, 2206648RP), 1× nonessential amino acids, 1× penicillin–streptomycin, 1× β-mercaptoethanol and 1,000 U ml$^{-1}$ LIF (homemade). Wild-type KH2, *Dnmt3a/b* DKO, *Dnmt* TKO (from Alexander Meissner Lab) and *Tet1/2/3* TKO ES cells (from Jacob Hanna Lab) were cultured on mitomycin-treated feeders produced in house. Wild-type E14s were acquired from Sarah Kinkley Lab, *Tet1/2* DKO cells were generated by us, and Tet1^Flag/Tet2^V5/Tet3^HA ES cells were gifted by Ian Chambers. Cell lines were not authenticated, but genotypes were verified by PCR. Cells tested negative for mycoplasma.

### In vitro diapause

ES cells or embryos were treated with the mTOR inhibitor INK128 or RapaLink-1 at 200 nM final concentration for the durations specified in individual experiments. To obtain embryos, 10- to 12-week-old b6d2F1 mice were superovulated via intraperitoneal injection with pregnant mare serum gonadotrophin (5 IU per 100 µl) on day 0, with human chorionic gonadotrophin 5 IU per 100 µl on day 2, and killed on day 3. Oocytes were collected and incubated with 10 µl of motile sperm in CARD medium (CosmoBio, KYD-003-EX) for in vitro fertilization. After overnight culture, two-cell-stage embryos were transferred to a fresh drop of K$^+$ simplex optimised medium (KSOM) (Merck, MR-107-D) and cultured until the blastocyst stage.

### Embryo transfer and in vivo diapause

Blastocysts were transferred into pseudopregnant females that have been previously mated with vasectomized males at E2.5. To induce diapause in vivo, ovariectomy was performed after embryo transfer as described previously[70]. Females were afterward injected every other day with 3 mg medroxyprogesterone 17-acetate subcutaneously. Diapaused blastocysts were flushed from uteri in M2 medium after 4 days of diapause at equivalent day of gestation (EDG) 7.5.

### Flow cytometry

Cells were dissociated from plates using TrypLE (Thermo Fisher 12604-021) and washed in Dulbecco's phosphate-buffered saline (Dulbecco). Cells were labeled with Alexa488-SSEA1 (BioLegend, 125610, 1:1,000) or Alexa647-AnnexinV (Invitrogen, A23204, 1:250) together with a live/dead cell stain excitable at 405 nm wavelength (Invitrogen, L34955), respectively, for 20 min in the dark on ice and subsequently washed in phosphate-buffered saline (PBS) containing 2% bovine serum albumin (BSA). After washing, fluorescence was measured on a BD FACSDIVA or BD FACSAriaII flow cytometer. Data analysis and visualization was performed using FlowJo (v10.8.2).

### Pharmacological treatment of embryos

Embryos were treated with Bobcat339 (100 µM final, Sigma SML2611), vitamin C (50 µM final, Sigma A4403), Rapalink (200 nM final Hölzel HY-111373) or α-ketoglutarate (200 µM final, Merck K1128) in KSOM medium (Merck, MR-107-D) in four-well dishes (Nunc IVF multidish, Thermo Scientific, 144444) in a volume of 500 µl. Recently, it has been shown that Bobcat339 effectivity is influenced by copper, which was not separated from the inhibitor stock used here[50]. Survival plots and statistical tests (log-rank test) were produced using RStudio (version 1.3.1093 with R version 3.6.3) with the survminer (version 0.4.9) package.

### Proliferation curves

Cells were seeded in six-well plates (Corning, 3516) at a density of 10$^5$ cells per well (on feeder cells when required, 2.5M per plate). INK128 (MedChemExpress, MCE-HY-13328) treatment was started the next day, and cells were counted every day using Cell Countess 3 (Invitrogen).

### Overexpression of wild-type or catalytically dead *Tet1* and/or *Tet2*

Wild-type *Tet1* or *Tet2* coding sequence was amplified from pcDNA3-Tet1 (Addgene 60938) and FH-Tet2-pEF (Addgene 41710) and cloned into a pCAGGS vector. To mutate catalytic activity, H1652Y and D1654A (*Tet1*) and H1304Y and D1306A (*Tet2*) were altered using the Q5 Site Directed Mutagenesis Kit (NEB, E0554S)[71]. Primer sequences:

(Tet1 F: 5′ GGCGATTCACAACATGCACAAC,
R: 3′ TTGTAAGAATGGGCACAAAAATC,
Tet2 F: 5′ AGCGCAGCAGAACATGCCAAATG,
R: 3′ CTGTAGGAATGAGCAGAGAAGTC).

### Generation of *Tet1/2* KO mouse ES cells

*Tet1* and *Tet2* genes were knocked out using gRNAs targeting *Tet1* exons 4 (GATTAATCACATCAACGCCG) and 13 (GCTTTGCGCTCCC-CCAAACGA) and *Tet2* exons 3 (GAGTGCTTCATGCAAATTCG) and 12 (GCTACACGGCAGCAGCTTCG). gRNA sequences were cloned into the pX330 plasmid with mCherry fluorescence. Wild-type E14 cells were nucleofected with the plasmids using the Lonza 4D Nucleofector. After 48 h, cells were single-cell sorted into 96-well plates using the BD FACSAria Fusion (Software v8.0.1). After 8 days, clones were screened using two primer pairs for each *Tet* gene.

Primer pair 1 (Tet1 F: 5′ AGCCATAGAAGCCCTGACTC,
R: 3′ CGGAGTTGAAATGGGCGAAA,
Tet2 F: 5′ CCGAAGCAACCGAACTCTTT,
R: 3′ ACAAGTGAGATCCTGGTGGG) binds outside the guide targeted region and only produces a PCR product after successful KO.
Primer pair 2 (Tet1 F: 5′ CGCCTGTACAAAGAGCTCAC,
R: 3′ AGGCTAGTCTCAGTTGGCAG;
Tet2 F: 5′ TTCTAATGCCTGTGTTCTCTCA,
R: 3′ CAACCTCTTTTGGCTCAGCT) binds at exon 6. KOs were confirmed via western blot using the TET1 (NBP2-19290, Novus Biologicals, 1:1,000) and TET2 (Cell Signaling Technology 45010S, 1:1,000) antibodies.

### IF imaging and quantifications

Cells and embryos fixed with 4% paraformaldehyde in Dulbecco's phosphate-buffered saline for 10 min. After fixation, cells were washed, permeabilized with 0.2% Triton X-100 in PBS for 10 min at room temperature, and blocked in 0.2% Triton X-100 containing 2% BSA and 5% goat serum for 1 h at room temperature. For 5mC and 5hmC IF, cells were depurinated after permeabilization using 2 N HCl for 1 h followed by a neutralization of 30 min in 0.1 M sodium borate. Cells were incubated with the following primary antibodies: anti-5mC (Diagenode, C15200003; 1:100), anti-5hmC (ActifMotif, 39769; 1:200) and anti-TFE3 (Merck, HPA023881; 1:50) overnight at 4 °C. Cells were washed and incubated with the following secondary antibodies for 1 h at room temperature: donkey anti-rabbit AF647 (Thermo Fisher, A32795, 1:1,000) and donkey anti-mouse AF488 (Thermo Fisher, A21202, 1:1,000).

Cells were mounted in Vectashield containing 4′,6-diamidino-2-phenylindole (DAPI; Vectashield, Cat: H-1200). Images were acquired on a ZEISS LSM880 microscope at 20× magnification, with Zen black and Zen blue software (version 2.3) and processed using Fiji (version 2.3.0) and CellProfiler[72] (version 4.2.1).

## Western blotting

Samples were mixed with 4× ROTI loading buffer (Carl Roth, K929.2), boiled at 98 °C for 5 min and loaded on 4–15% Mini-PROTEAN®TGX precast protein gels (Bio-Rad, 4561083). Proteins were separated by electrophoresis at 70 V for 15 min followed by 100 V for 1 h using 10× Tris/glycine/sodium dodecyl sulfate running buffer (Bio-Rad, 1610772). Proteins were transferred to a polyvinylidene fluoride membrane (Thermo Fisher Scientific, IB24001) using the iBlot 2 dry blotting system (Thermo Fisher Scientific, IB21001) and run at 20 V for 7 min. Membranes were blocked with 5% milk in TBS-T buffer (Thermo Fisher Scientific, 28360) for 1 h at room temperature, and incubated with primary antibody (indicated in each method section) in 5% milk in TBS-T buffer overnight, followed by secondary antibody at room temperature for 1 h. For detection, membranes were incubated with ECL Western Blotting Substrate (Thermo Fisher Scientific, 32106) for 1 min before imaging with the ChemiDoc system (Bio-Rad).

## RNA-seq

Cells were trypsinized and sorted on a BD FACSAria Fusion (Software v8.0.1 configuration 2B-5YG-3R-2UV-6V). Total RNA was extracted from 200,000 cells using the Qiagen RNeasy kit (Qiagen, 74004). External RNA Controls Consortium (ERCC)[68] RNA Spike-In Mix (Thermo, 4456740) was used. Libraries were prepared from 500 ng total RNA using KAPA RNA HyperPrep Kit with RiboErase (Roche, 8098131702) following the manufacturer's instructions, and sequenced on a NovaSeq 600, S4 flow cell, paired-end mode. Raw reads were subjected to adapter and quality trimming with cutadapt[73] (version 2.4; parameters: –quality-cutoff 20–overlap 5–minimum-length 25–interleaved–adapter AGATCGGAA-GAGC-A AGATCGGAAGAGC), followed by poly(A) trimming (parameters: –interleaved–overlap 20–minimum-length–adapter "A[100]"–adapter "T[100]"). Reads were aligned to the mouse reference genome (mm10) using STAR (version 2.7.5a; parameters: –runMode alignReads–chimSegmentMin 20–outSAMstrandField intronMotif–quantMode Gene-Counts)[74], and transcripts were quantified using stringtie (version 2.0.6; parameters: -e)[75] with GENCODE annotation (release VM19). For the repeat expression quantification, reads were realigned with additional parameters '–outFilterMultimapNmax 50'. Differential gene expression analysis was performed on stringtie output using DEseq2 (ref. [76]) (version 1.38.2).

## ATAC-seq

A total of 50,000 cells per sample were collected as described above. The ATAC-seq protocol from Corces et al.[77] was followed. Illumina transposase was used (20034198). Samples were purified using the Zymo DNA Clean and Concentrator-5 Kit (D4014). Libraries were amplified with eight PCR cycles with i5 and i7 primers from ref. [78]. The final number of cycles was determined following ref. [78]. Libraries were sequenced as above. Raw reads were subjected to adapter and quality trimming with cutadapt as above and aligned to the mouse genome (mm10) using BWA with the 'mem' command (version 0.7.17, default parameters)[79]. A sorted binary alignment map (BAM) file was obtained and indexed using SAMtools with the 'sort' and 'index' commands (version 1.10)[80]. Duplicate reads were identified and removed using GATK (version 4.1.4.1) 'MarkDuplicates' and default parameters. Replicates were merged using SAMtools 'merge'. Peaks were called using the MACS2 (ref. [81]) peakcall (2.1.2_dev) function with default parameters.

## Whole genome bisulfite sequencing

A total of 50,000 cells per sample were collected as described above. Genomic DNA was isolated using the Purelink Genomic DNA mini kit (Invitrogen). Libraries were prepared using the Accel-NGS Methyl-Seq DNA Library Kit. Libraries were sequenced as above. Raw reads were subjected to adapter and quality trimming using cutadapt as above (Illumina TruSeq adapter clipped from both reads), followed by trimming of 10 and 5 nucleotides from the 5′ and 3′ end of the first read and 15 and 5 nucleotides from the 5′ and 3′ end of the second read[73].

Trimmed reads were aligned to the mouse genome (mm10) using BSMAP (version 2.90; parameters: -v 0.1 -s 16 -q 20 -w 100 -S 1 -u -R)[82]. A sorted BAM file was obtained and indexed using samtools with the 'sort' and 'index' commands (version 1.10)[80]. Duplicates were removed using the 'MarkDuplicates' command from GATK (version 4.1.4.1) and default parameters[83]. Methylation rates were called using mcall from the MOABS package (version 1.3.2; default parameters)[84]. All analyses were restricted to autosomes, and only CpGs covered by at least 10 and at most 150 reads were considered for downstream analyses.

## CUT&Tag

CUT&Tag was performed as described previously in ref. [85]. A total of $10^5$ nuclei were incubated with the following primary antibodies overnight at 4 °C: TET1, NBP2-19290, Novus Biologicals, 1:100; TET2, Cell Signaling Technology 45010S, 1:50; TFE3, Sigma HPA023881, 1:50; IgG, Abcam ab46540, 1:100; H3K27ac, 9733S, CST, 1:100; H3K4me3, 9751S, CST, 1:100; H3K4me1, 5326S,CST, 1:100. Guinea pig α-rabbit secondary antibody (ABIN101961, Antibodies Online, 1:100) was used. For tagmentation, homemade 3xFLAG-pA-Tn5 preloaded with Mosaic-end adapters was used. DNA was purified using Chimmun DNA Clean & Concentrator (D5205, Zymo Research).

Libraries were amplified using the NEBNext HiFi 2× PCR Master Mix (New England BioLabs) with i5- and i7-barcoded primers[78] and cleaned up using Ampure XP beads (Beckman Coulter). Library quality control was done using the Agilent High Sensitivity D5000 Screen-Tape System and Qubit dsDNA HS Assay (Invitrogen). Libraries were sequenced as above. Raw reads were trimmed using cutadapt (version 2.4; parameters: –quality-cutoff 20–overlap 5–minimum-length 25–adapter AGATCGGAAGAGC -A AGATCGGAAGAGC) and aligned to the mouse genome (mm10) using BWA with the 'mem' command (version 0.7.17, default parameters)[79]. A sorted BAM file was obtained and indexed using samtools with the 'sort' and 'index' commands (version 1.10)[80]. Duplicate reads were identified and removed using GATK (version 4.1.4.1) 'MarkDuplicates' and default parameters. Relicates were merged using SAMtools 'merge'. Peaks were called using the MACS2 (ref. [81]) peakcall (2.1.2_dev) with default parameters.

## Pathway expression analysis

Pathway expression value was defined as the mean expression (transcripts per million, TPM) of genes in a given pathway at the indicated time points. Kyoto Encyclopedia of Genes and Genomes[86] pathways containing at least ten genes were included in the analysis.

## Motif enrichment analysis

Motif enrichment was performed using Homer (v4.7, 8-25-2014). TET dormancy targets were compared against the mouse genome (mm10) using the '-size given' setting.

## TF footprinting analysis

ATAC-seq peaks were called by MACS2 (2.2.7.1) with 75-bp shift and 150-bp extension. Differential TF footprints inside these peak regions were identified by TOBIAS[36] (0.12.11) using Homer motifs (v4.7, 8-25-2014).

## Definition of mouse ES cell enhancer sets

Active and primed enhancer were retrieved from ref. [87] and are defined as follows: active enhancers, genomic regions with p300 enrichment, located within 1 kb of regions enriched in H3K27ac and not enriched in H3K27me3 (within 1 kb); primed enhancers, genomic regions with H3K4me1 enrichment and not enriched in H3K27me3 or H3K27ac (within 1 kb).

## Generation of *Tet* KO embryos via Cas9-assisted gene editing

In vitro-fertilized zygotes were electroporated to generate KOs, as previously described[88]. In brief, oocytes from superovulated B6D2F1

female mice (7–9 weeks old; Envigo) and sperm from F1B6xCAST was incubated for in vitro fertilization, as previously described[89]. Pronuclei stage 3 zygotes were rinsed with M2 (Sigma) and OptimMEM I (Gibco, 31985062) medium before electroporation. Three gRNAs per gene were designed targeting the first few exons. gRNAs were assembled with CAS9 into ribonucleoproteins[88]. Embryos were electroporated on a NEPA21 (Nepagene) in a chamber with 5 mm electrode gap and the following settings: four poring pulses with a voltage of 225 V, pulse length of 2 ms, pulse interval of 50 ms, decay rate of 10% and uniform polarity, followed by five transfer pulses with a voltage of 20 V, pulse length of 50 ms, pulse interval of 50 ms, decay rate of 40% and alternating polarity. Electroporated zygotes were rinsed in KSOM drops (Merck, MR-106-D) and cultured until blastocyst stage. In the case of embryo transfer, 15 blastocysts were transferred into each uterine horn of pseudopregnant female CD-1 (21–25 g, Envigo, age 7–12 weeks) mice 2.5 days post-coitum. E8.5-stage embryos were isolated from the uteri of foster mice. The embryos were dissected in 1× Hanks' Balanced Salt Solution (Gibco) on ice after the decicuda were removed. Embryos were washed in 1× PBS (Gibco) with 0.4% BSA and imaged on an Axiozoom (ZEISS) microscope. Images were processed with Fiji.

For KO validation qPCRs, RNA was isolated using Arcturus Pico Pure RNA isolation kit (Biosystems) and reverse transcribed using High-Capacity cDNA synthesis kit (KAPA biosystems). RT–qPCR was done using Kapa SYBR 2× master mix. β-Actin was used for normalization. qPCR results were visualized using GraphPad Prism v10.

gRNA sequences:
*Tet1*:

1. TCGATCCCGATTCATTCGGG
2. TTGGCGGCGTAGAATTACAT
3. GATTAATCACATCAACGCCG

*Tet2*:

1. AAGATTGTGCTAATGCCTAA
2. GAGTGCTTCATGCAAATTCG
3. GCTCCTAGATGGGTATAATA

*Tet3*:

1. GAGCGCGCTGAGCATTGCCA
2. TTCTATCCGGGAACTCATGG
3. TCGGATTGTCTCCCGTGAGG

Control guide against green fluorescent protein:
GAAGTTCGAGGGCGACACCC

qPCR primers
Tet1:
F: 5′ ACCACAATAAATCAGGTTCACAC, R: 5′ TCTCCACTGCACAATGCCTT
Tet2:
F: 5′ GCAATCACCACCCAGTAGAA, R: 5′ TCCACGTGCTGCCTATGTAC
Tet3:
F: 5′ GCCTCAATGATGACCGGACC, R: 5′ ATGAGTTTGGCAGCGAGGAA
β-Actin:
F: 5′ TGGGTGTATTCCAGGGAGAG, R: 5′ AAGGCCAACCGTGAAAAGAT

## FLASH

*Tet1* and *Tet2* genes were endogenously tagged with the biotinylation signal necessary to perform the FLASH method using a modified plasmid from the CRISpaint toolkit[90], which contains a biotinylation signal followed by FLAG tag and a puromycin resistance gene as the donor plasmid. gRNAs targeting the C-terminus of *Tet1* (GTTGCGGGACCCTACAATCGT) and *Tet2* (GACAACACATTTGTATGACGC), respectively, were expressed from the px330 plasmid together with *Cas9*. pX330 expressing the appropriate gRNA together with the donor plasmid and

the appropriate frame selector plasmid were nucleofected to generate each cell line. Colonies were delected using Puromycin for 10 days. The following primers were used for genotyping:

*Tet1*:
F: 5′ GGGAGTGTCCTGATGTATCCCCCG
R: 3′ CTCAGCTCATCACTCCGTGTGTTGA

*Tet2*:
F: 5′ CCAGTCTCTTGCTGAGAACACAGGG
R: 3′ CAGATGCTGTGACCTGTCCCTACG.

Successful knock-in of endogenous *Tet1* and *Tet2* was confirmed by a streptavidin pulldown followed by western blotting. Membranes were incubated with horseradish peroxidase-conjugated streptavidin (Thermo, N100) to visualize biotinylated TET1 and TET2 proteins.

FLASH was performed following the protocol in ref. 46. Cells were crosslinked with 0.15 mJ cm$^{-2}$ ultraviolet (UV)-C irradiation. Isolated RNA was reverse-transcribed and RNase H-treated. cDNA was column-purified and circularized with CircLigase for 2–16 h. Circularized cDNA was directly PCR amplified, quantified and sequenced on Illumina NextSeq 500 in paired-end mode.

## IP–MS

Cells were fractionated to collect nuclei. For this, cells were lysed using cold buffer A (10 mM HEPES pH 7.9, 5 mM MgCl$_2$, 0.25 M sucrose and 0.1% NP-40), incubated for 10 min on ice and passed four times through an 18G1 needle (BD Microlance 3, 304622), and centrifuged. The pelleted nuclei was lysed in cold buffer B (10 mM HEPES pH 7.9, 1 mM MgCl$_2$, 0.1 mM EDTA, 25% glycerol and 0,5 M NaCl), incubated for 30 min on ice, passed four times through an 18G1 needle and sonicated using Bioruptor 300 (Diagenode) with settings 30 s on, 30 s off for 5 min at 4 °C. Protein concentration was quantified using the BCA Protein Assay Kit (Pierce, 23225). One milligram of protein was used per pulldown in a minimum volume of 500 µl in IP buffer (20 mM HEPES pH 7.9, 25% glycerol, 0.15 M NaCl, 1.5 mM MgCl$_2$, 0.2 mM EDTA and 0.02% NP-40). FLAG antibody (Merck, F3165) and Dynabeads Protein A (Invitrogen, 10001D) were added (1 mg of beads and ~8 µg of antibody) and the mixture incubated for 2 h at 4 °C. After pulldown, beads were washed 3× in IP buffer and immediately digested for MS or boiled at 96 °C in 30 µl of Leammli buffer (Roth, K930.1) for western blotting.

**MS.** A total of 135 µl of 100 mM ammonium bicarbonate was added to the washed magnetic beads. This was followed by a tryptic digest including reduction and alkylation of the cysteines. The reduction was performed by adding tris(2-carboxyethyl)phosphine with a final concentration of 5.5 mM at 37 °C on a rocking platform (700 rpm) for 30 min. For alkylation, chloroacetamide was added with a final concentration of 24 mM at room temperature on a rocking platform (700 rpm) for 30 min. Then, proteins were digested with 200 ng trypsin (Roche) per sample, shaking at 1,000 rpm at 37 °C for 18 h. Samples were acidified by adding 6 µl 100% formic acid (2% final), centrifuged shortly and placed on the magnetic rack. The supernatants, containing the digested peptides, were transferred to a new low-protein-binding tube. Peptide desalting was performed on C18 columns (Pierce). Eluates were lyophilized and reconstituted in 11 µl of 5% acetonitrile and 2% formic acid in water, briefly vortexed and sonicated in a water bath for 30 s before injection to nano-liquid chromatography (LC)–tandem MS.

**Run parameters.** LC–tandem MS was carried out by nanoflow reverse-phase LC (Dionex Ultimate 3000, Thermo Scientific) coupled online to a Q-Exactive HF Orbitrap mass spectrometer (Thermo Scientific), as reported previously. Briefly, the LC separation was performed using a PicoFrit analytical column (75 µm inner diameter × 50 cm long, 15 µm Tip inner diameter; New Objectives) in-house packed with 3-µm C18 resin (Reprosil-AQ Pur, Dr. Maisch).

**Peptide analysis.** Raw MS data were processed with MaxQuant software (v1.6.10.43) and searched against the mouse proteome database UniProtKB with 55,153 entries, released in August 2019.

## Enhancer and L1Md gene contact analysis

To determine the genes looping with the identified active enhancers and potentially by L1Md elements, we used the publicly available predictions from the Activity-By-Contact model[49]. For gene looping with enhancers, we used the recommended cut off (ABC score 0.02). For potential L1Md–promoter contacts, the HiC contact probability (>15) provided in the same file for mouse ES cells was used (mESC.AllPredictions.txt).

## Gene knockdowns

Three short hairpin RNAs (shRNAs) against TFE3 were cloned into a pLKO.1 plasmid containing a puromycin resistance gene and allowing for doxycycline-inducible expression, respectively, resulting in pLKO.1.shTFE3:

shTFE3_1: ATCCGGGATTGTTGCTGATAT, shTFE3_2: GTGGATTA-CATCCGCAAATTA, shTFE3_3: AGCTATCACCGTCAGCAATTC.

Two micrograms of pLKO.1.shTFE3 was co-transfected with 2 μg of equal parts pVSV-G, pMDL and pRSV (packaging vectors) into HEK 293T cells grown to 80% confluency on a 10-cm uncoated culture dish. After 24 h, cell culture supernatant was collected for 3 consecutive days to enrich for produced viruses. The virus supernatant was concentrated and used to transduce E14 wild-type cells. For transduction, 100,000 E14 cells were mixed with 50 μl of concentrated virus suspension and 10 μg ml$^{-1}$ polybrene in a 1.5 ml tube and rotated at 37 °C for 1 h. Subsequently, cells were plated on six-well culture dishes and grown for 48 h after which puromycin selection was applied for 6 days. Efficient knockdown of TFE3 was confirmed by IF followed by confocal microscopy.

## Reporting summary

Further information on research design is available in the Nature Portfolio Reporting Summary linked to this article.

## Data availability

RNA-seq, whole genome bisulfite sequencing, ATAC-seq, CUT&TAG and FLASH datasets generated in this study have been deposited to the GEO database under the accession number GSE221470 (https://www.ncbi.nlm.nih.gov/geo/query/acc.cgi?acc=GSE221470). IP–MS data have been deposited to the PRIDE[91] depository with the identifier number PXD039056. Source data are provided with this paper.

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

## Acknowledgements

We thank D. Iyer, N. Lopez Anguita, C. Haggerty, D. Hnisz and M. Ramalho-Santos for discussions and feedback; scientific facilities of the Max Planck Institute for Molecular Genetics for excellent service and discussions; J. Shay, M. Walther, B. Romberg and C. Mancini for assistance. We thank S. Haas, P. Guckelberger, M. Robson and D. Ibrahim for help with HiC analysis, J. Hanna for providing *Tet1/2/3* flox/flox ES cells, and I. Chambers for providing Tet1$^{Flag}$/Tet2$^{V5}$/Tet3$^{HA}$ ES cells. This project was supported by the Max Planck Society (T.A., H.K., M.V., A.M. and A.B.-K.), a Swiss National Science Foundation Early Postdoc.Mobility fellowship P2EZP3_195682 (V.A.v.d.W.), the Deutsche Forschungsgemeinschaft (DFG) international training group IRTG2403 (Y.Z.), and the Sofja Kovalevskaja Award (Humboldt Foundation) to A.B.-K.

## Author contributions

A.B.-K. conceived the project. M.S. generated the *Tet1/2* DKO ES cells and performed all experiments and data analyses except the following: C.-Y.C. performed CUT&Tag experiments and initial data analyses, I.A.I. taught the FLASH experiment and performed data analyses, A.S.K. helped with the generation of *Tet* TKO embryos, V.A.v.d.W. performed pathway expression analysis, Y.Z. performed TF footprinting analysis, H.K. performed and supervised computational analyses, and P.A.O. helped with CUT&Tag analyses. T.A., M.V. and A.M. gave feedback, and A.B.-K. supervised the project. M.S. and A.B.-K. wrote the paper with feedback from authors.

## Funding

## Competing interests

I.A.I. and T.A. are inventors on a patent application (no. EP3325621B1, European Patent Office) regarding the s-oligo design used in FLASH experiments. The other authors declare no competing interests.

## Additional information

**Extended data** is available for this paper at https://doi.org/10.1038/s41594-024-01313-7.

**Correspondence and requests for materials** should be addressed to Aydan Bulut-Karslioğlu.

**a**

| | | wild-type | | | | Tet DKO | | |
|---|---|---|---|---|---|---|---|---|
| **ATAC-seq** | | 0h | 24h | 72h | 144h | 0h | 24h | 72h |
| read (M) | rep1 | 86 | 76.3 | 105.8 | 89.2 | 87.8 | 79.9 | 82.3 |
| | rep2 | 73.5 | 95.1 | 42.9 | 99 | 117.7 | 81.1 | 92.8 |
| % unique | rep1 | 78.4 | 77.9 | 74.7 | 76.1 | 76 | 75.9 | 78.5 |
| | rep2 | 72.1 | 69.5 | 76.7 | 73.6 | 67.6 | 73.1 | 75.5 |
| **RNA-seq** | | | | | | | | |
| read (M) | rep1 | 86 | 81.2 | 82.3 | 63.2 | 66.9 | 66.8 | 75.6 |
| | rep2 | 70.4 | 64.9 | 85.4 | 62.9 | 72.8 | 79 | 78.6 |
| % unique | rep1 | 81.32 | 84.08 | 83.47 | 84 | 83.27 | 84.26 | 84.61 |
| | rep2 | 82.8 | 83.18 | 82.75 | 84.4 | 84.94 | 84.22 | 85.3 |
| **WGBS** | | | | | | | | |
| read (M) | rep1 | 788200524 | 677440460 | 681144072 | 683895094 | 862772162 | 691244532 | 632944502 |
| | rep2 | 850334502 | 880940066 | 756842002 | 684162458 | 868365548 | 621193401 | 762636108 |
| % unique | rep1 | 75.3683 | 76.3289 | 77.9778 | 76.2147 | 75.5012 | 77.0858 | 78.7511 |
| | rep2 | 75.1371 | 76.0342 | 76.3423 | 77.6797 | 73.8008 | 78.2541 | 76.9717 |
| Bisulfite conversion rate | rep1 | 0.994546 | 0.994378 | 0.994378 | 0.991211 | 0.995621 | 0.994872 | 0.99394 |
| | rep2 | 0.99505 | 0.992665 | 0.993836 | 0.992665 | 0.995798 | 0.9951 | 0.995127 |

**b**                                                                                                **c**

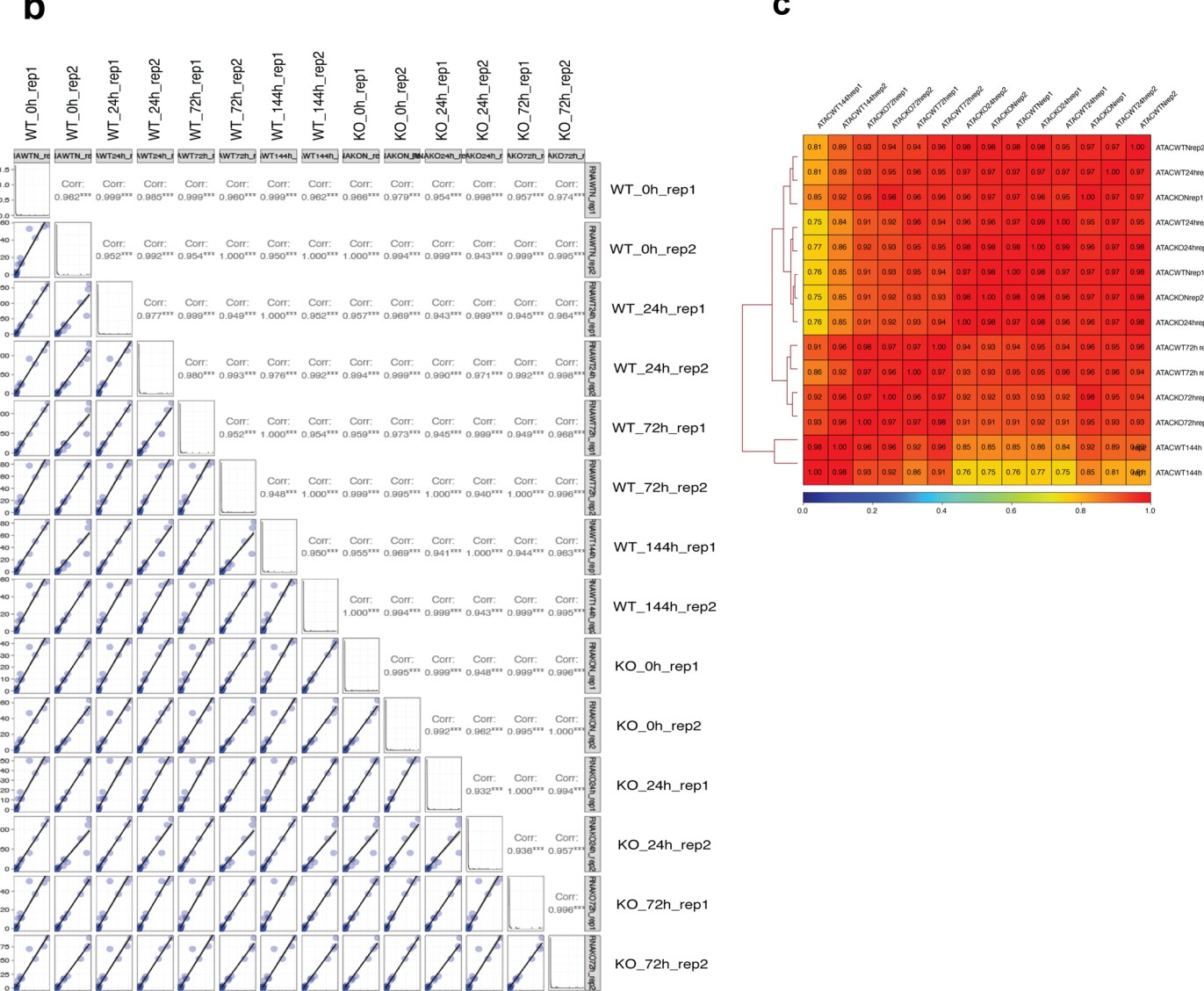

**Extended Data Fig. 1 | Further characterization of genomic features of paused pluripotency. a**. Summary of read depth and quality control parameters of RNA-seq, WGBS, and ATAC-seq experiments. **b**. Plots showing the linearity of ERCC spike-ins in all RNA-seq samples. Correction factor was extracted by calculating the ratio of reads mapped to ERCCs and the mouse genome. **c**. Global clustering of ATAC-seq samples.

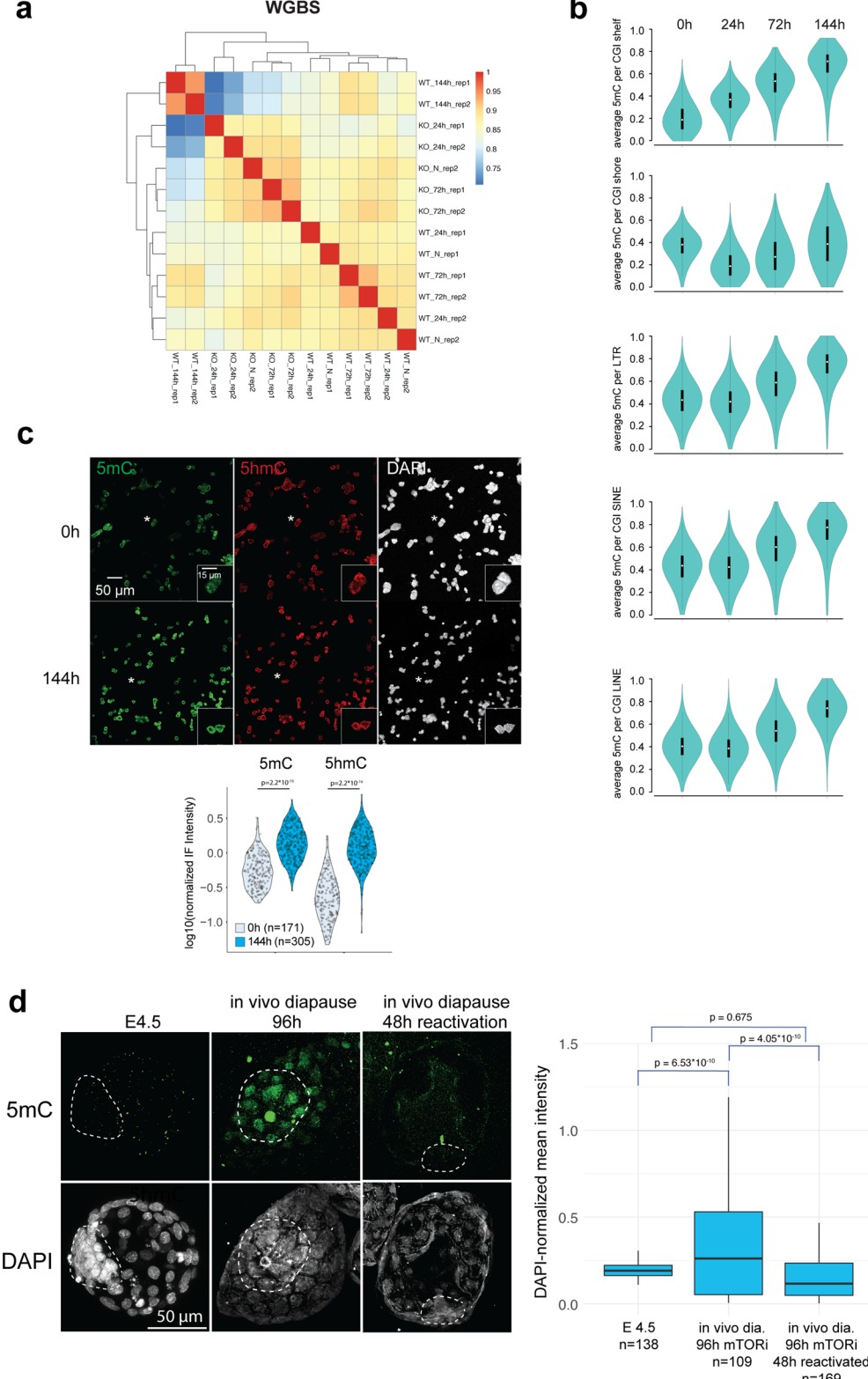

**Extended Data Fig. 2 | DNA methylation dynamics during diapause and reactivation. a.** Global clustering of WGBS samples. **b.** DNA methylation levels of different genomic features in wild-type ESCs over time during mTORi treatment. White dots show the median; vertical box plots within violin plots show interquartile range (IQR) and the whiskers show 1.5 IQR. **c.** Immunofluorescence of untreated or mTORi treated ESCs for 5mC and 5hmC methylation. Bottom, single-nucleus quantifications of 5mC and 5hmC intensities normalized to DAPI. n, number of cells. Statistical test is two-tailed t-test. **d.** Immunofluorescence of E4.5, in vivo diapaused, and reactivated (48 h post in vivo diapause) mouse blastocysts for 5mC methylation. Right, single-nucleus quantifications of 5mC intensity normalized to DAPI. Horizontal line shows the median, box spans the IQR and whiskers span 1.5 IQR. n, number of cells. Statistical test is a one-way Anova with Tukey's multiple comparison test. Dashed lines mark the ICM.

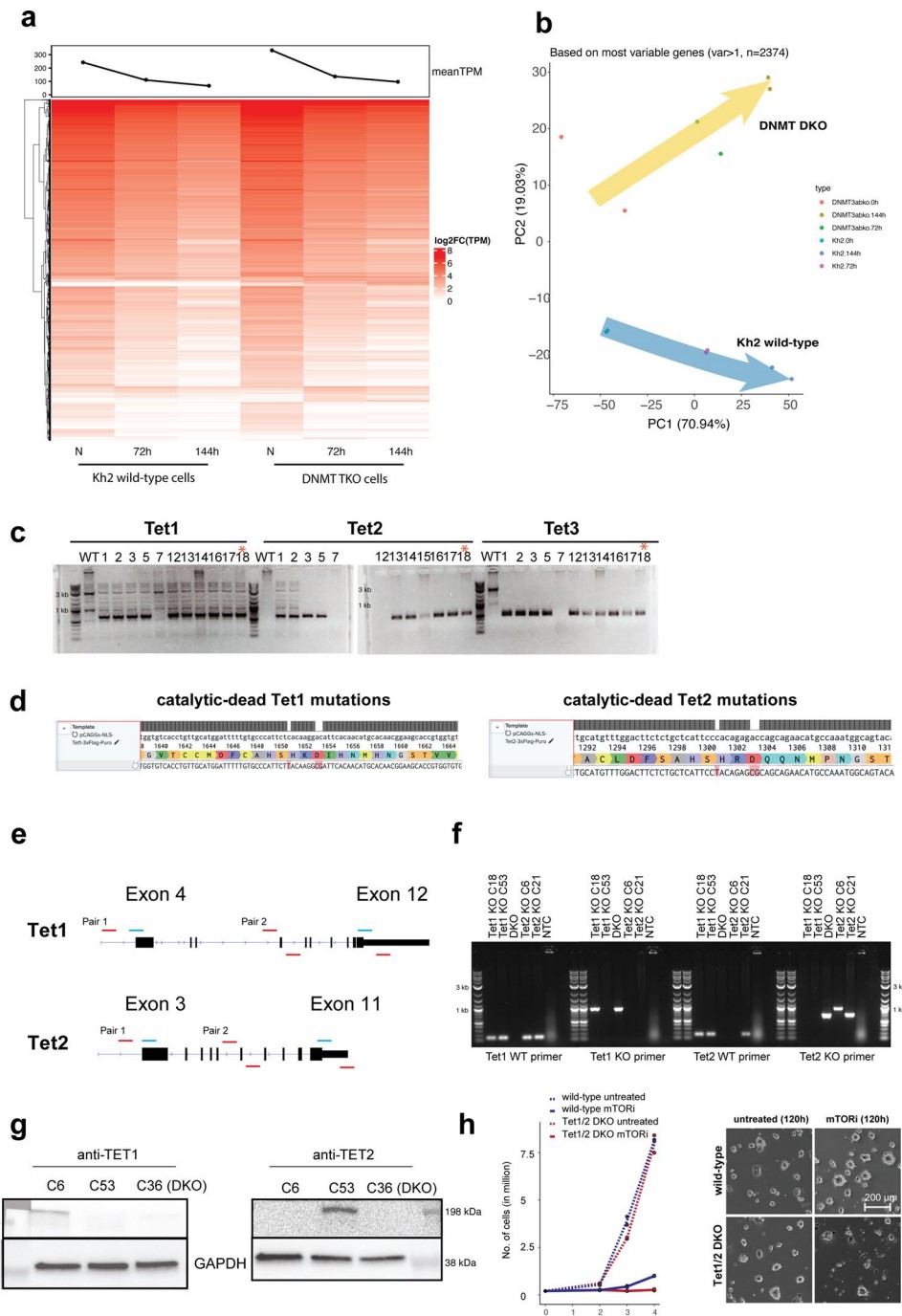

**Extended Data Fig. 3 | Characterization of *Tet* KO and *Dnmt* KO ESC. a.** Bulk RNA-seq heatmap showing expression of all genes over time in wild-type vs *Dnmt3a/b* DKO ESCs treated with mTORi. All samples are normalized to ERCC spike-in RNAs to accurately reflect global changes. Line plot on top shows mean TPM at each time point. **b.** Principal components analysis (based on 5000 most variable genes between 0 h and 144 h wild-type) of RNA-seq samples. Two replicates were performed for all experiments. **c.** Genomic PCR of *Tet1/2/3* TKO iPSCs. Parental *Tet1/2/3*^flox/flox cells were transiently transfected with a plasmid carrying the Cre recombinase. Single cells were sorted on 96-well plates and individual clones were genotyped. Clone 18 (indicated with asterisks) was used

for follow-up experiments. **d.** DNA sequence of catalytic-dead (cd) Tet1 and Tet2 overexpression constructs. Mutated sequences are highlighted in red. **e.** Strategy for generating *Tet1/2* DKO ESCs. Blue lines indicate gRNAs and red lines indicate PCR primers. **f.** Genotyping PCR of *Tet1/2* DKO ESCs with indicated primers. **g.** Western blot results showing depletion of TET1 and TET2 in *Tet1/2* DKO (C36) ESCs. **h.** Bright field images and survival curves of wild-type and *Tet1/2* DKO ESCs (E14 cells, feeder-independent) treated with the mTOR inhibitor INK128. Images are representative for two biological replicates. Individual data points shown, lines denote the mean.

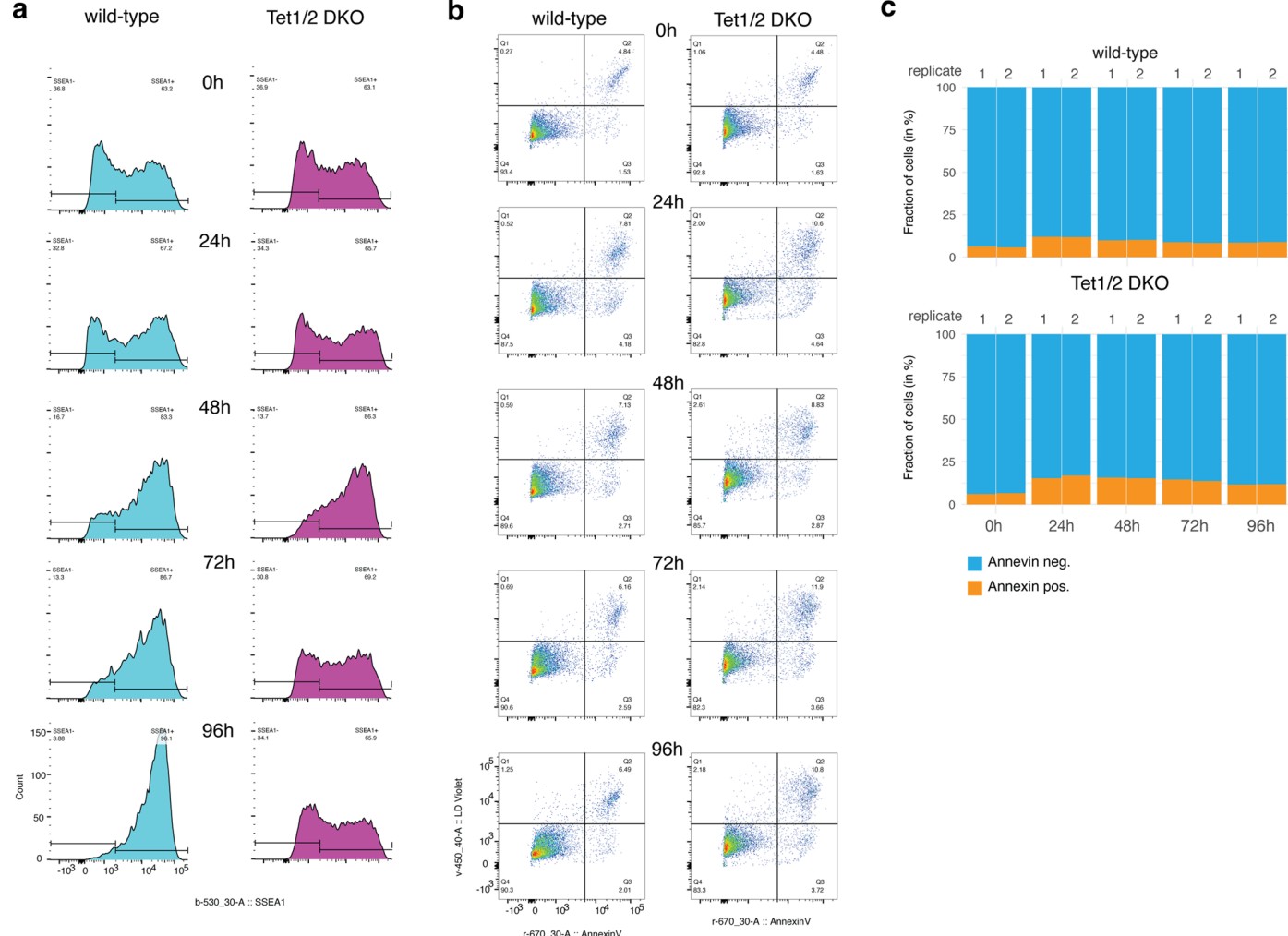

**Extended Data Fig. 4 | Detailed characterization of wild-type and *Tet1/2* DKO ESCs' response to mTORi-induced dormancy. a**. Flow cytometry analysis of SSEA1 expression levels (a pluripotency marker) in wild-type and *Tet1/2* DKO cells in normal and mTORi conditions. **b**. Flow cytometry analysis of apoptosis levels in wild-type and *Tet1/2* DKO cells in normal and mTORi conditions. Annexin V was used as an apoptosis marker. **c**. Quantification of Annexin V-positive and -negative cells at each time point. Data is shown for two biological replicates.

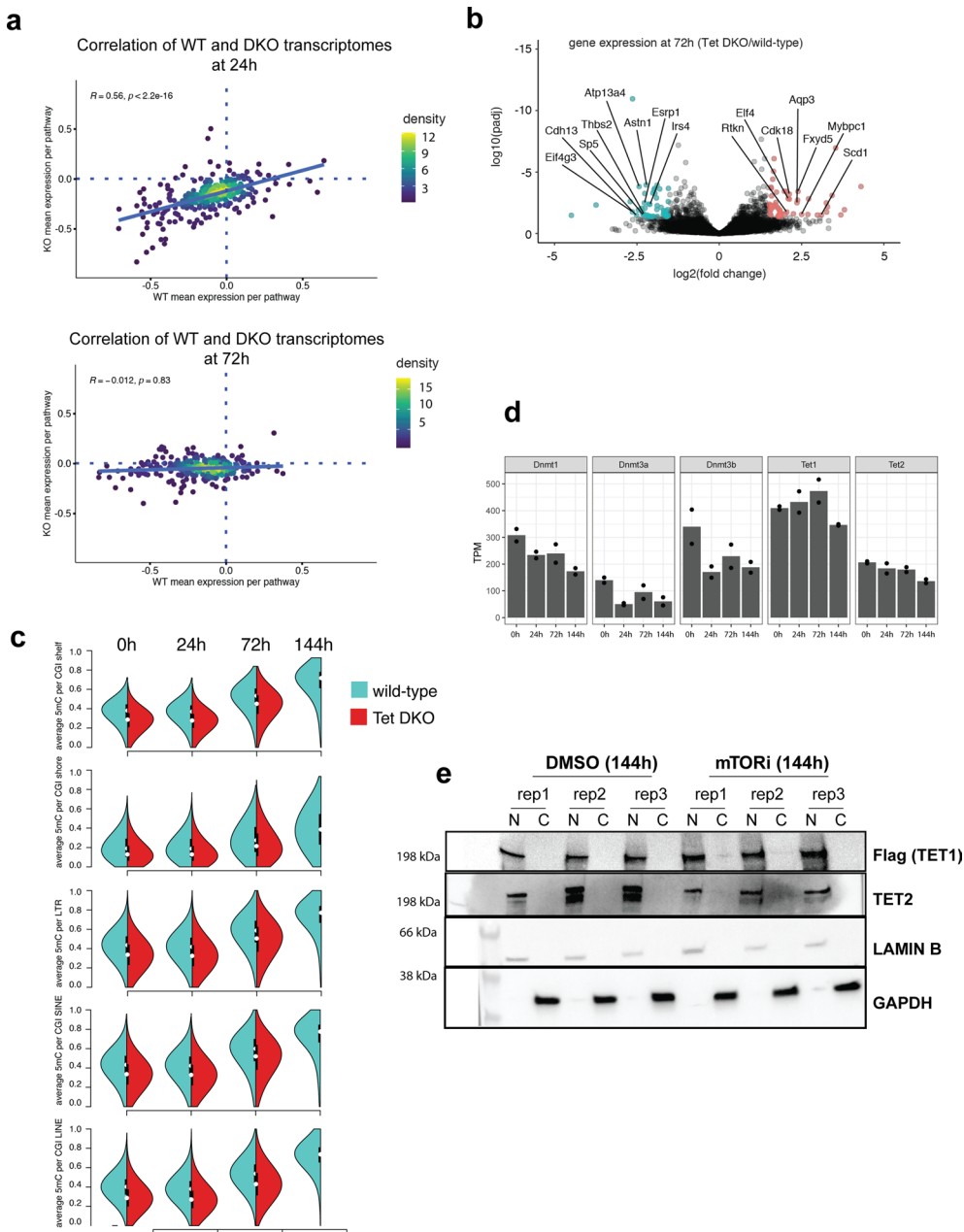

**Extended Data Fig. 5 | Further characterization of *Tet1/2* DKO ESCs' response to mTORi-induced dormancy. a**. Correlation of pathway expression in wild-type and *Tet1/2* DKO ESCs at 24 h and at 72 h of mTORi treatment. Significance of correlation was tested by a linear regression t-test (two-tailed). **b**. Differentially expressed genes in *Tet1/2* DKO vs. wild-type at 72 h of mTORi treatment. P-values and log2 fold changes were derived using DEseq2. **c**. DNA methylation levels of different genomic features in wild-type vs *Tet1/2* DKO ESCs. White dots show the

median; vertical box plots within violin plots show interquartile range (IQR) and the whiskers show 1.5 IQR. **d**. RNA expression levels of Tet and Dnmt enzymes in untreated vs mTORi treated wild-type ESCs over time. TPM, transcripts per kilobase million. **e**. Expression of TET1 and TET2 proteins in untreated vs mTORi treated (144 h) wild-type ESCs. N: nuclear, C: cytoplasmic extract. Three biological replicates were performed.

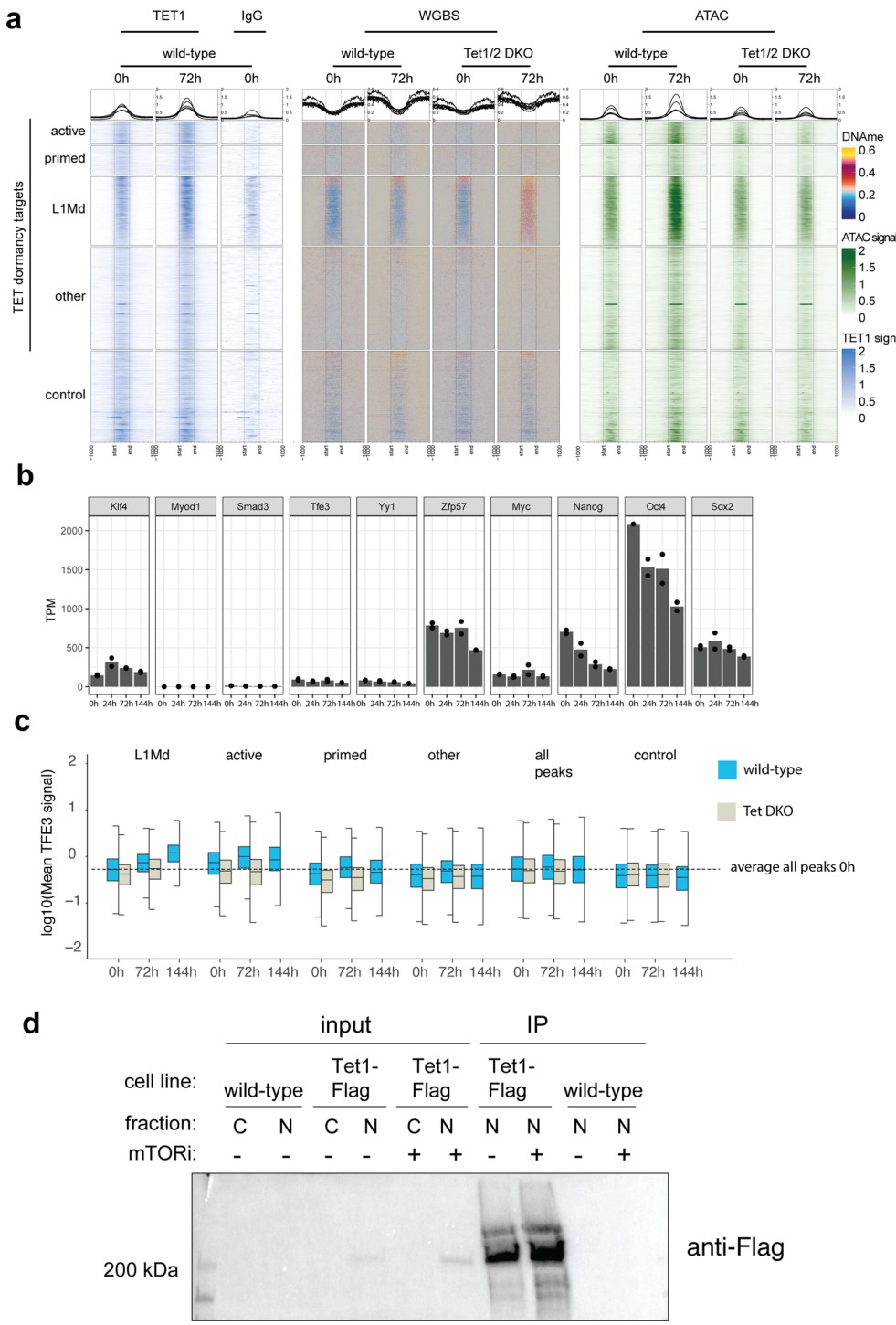

**Extended Data Fig. 6 | Further analysis of TFE3 binding levels at TET-target and control regions. a**. Heatmaps showing TET occupancy (TET1), DNA methylation (WGBS), and genome accessibility (ATAC-Seq) at TET-dormancy-targets vs control regions. **b**. RNA expression levels (TPM) of the transcription factors shown in Fig. 5a and b. **c**. Boxplots showing TFE3 binding levels at TET-dormancy-targets and control regions. Horizontal lines denote the median, lower and upper hinges denote the first and third quartiles. Whiskers show 1.5 times the interquartile range. **d**. Western blot showing Flag-immunoprecipitated material from Tet1-Flag and wild-type ESCs. Three biological replicates were performed.

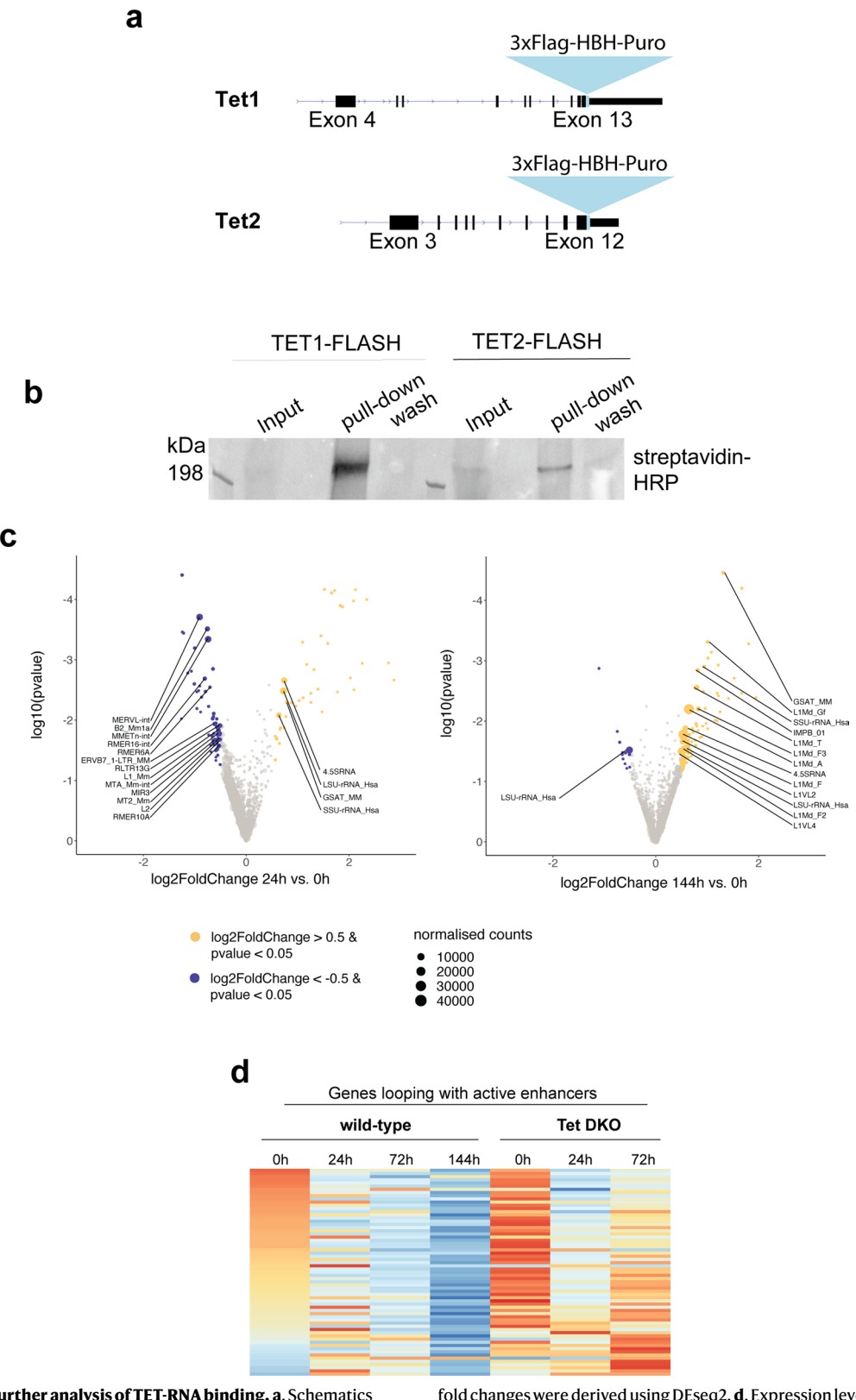

**Extended Data Fig. 7 | Further analysis of TET-RNA binding. a**. Schematics of FLASH tag insertion into the *Tet1* and *Tet2* loci. **b**. Western blots showing successful pull-down of the tagged TET1 and TET2 proteins. **c**. Differential binding levels of TET1 to repeat RNAs at 24 h and 144 h compared to 0 h. Gray indicates no significant differences between the time points. P-values and log2 fold changes were derived using DEseq2. **d**. Expression levels of top genes that contact TET-target active enhancers (ABC-score[49]>0.02) as determined by analysis of published HiC and H3K27ac datasets. TET activity at enhancers is largely uncoupled from transcription under dormancy conditions.

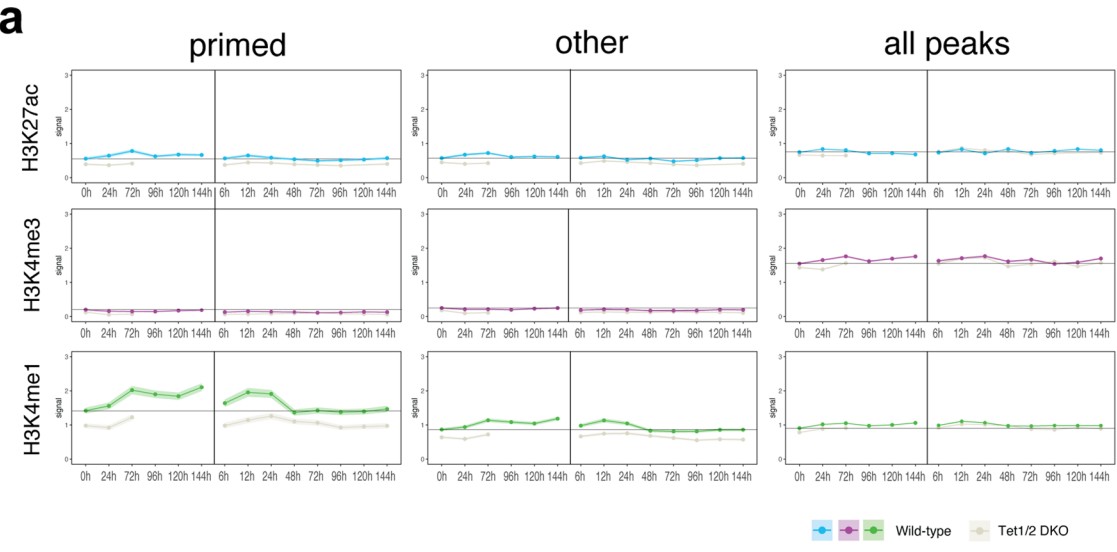

**Extended Data Fig. 8 | See next page for caption.**

**Extended Data Fig. 8 | Further investigation of chromatin dynamics of pausing. a**. Levels of the indicated histone marks at TET-target primed enhancers, the Other category (see Fig. 4c), and at all TET-bound peaks in wild-type and *Tet1/2* DKO cells over 144 h time course of mTORi-mediated pausing and release. *Tet1/2* DKO cells were paused for 72 h to avoid loss of pluripotent colonies. Lines show mean values, shade shows the confidence interval. Dashed lines denote levels of each mark at 0 h in each genetic background. **b**. Levels of the indicated histone marks at the shown regions during longer-term pausing. Time points up to 15 days were collected in wild-type cells. Lines show mean values, shade shows the confidence interval.

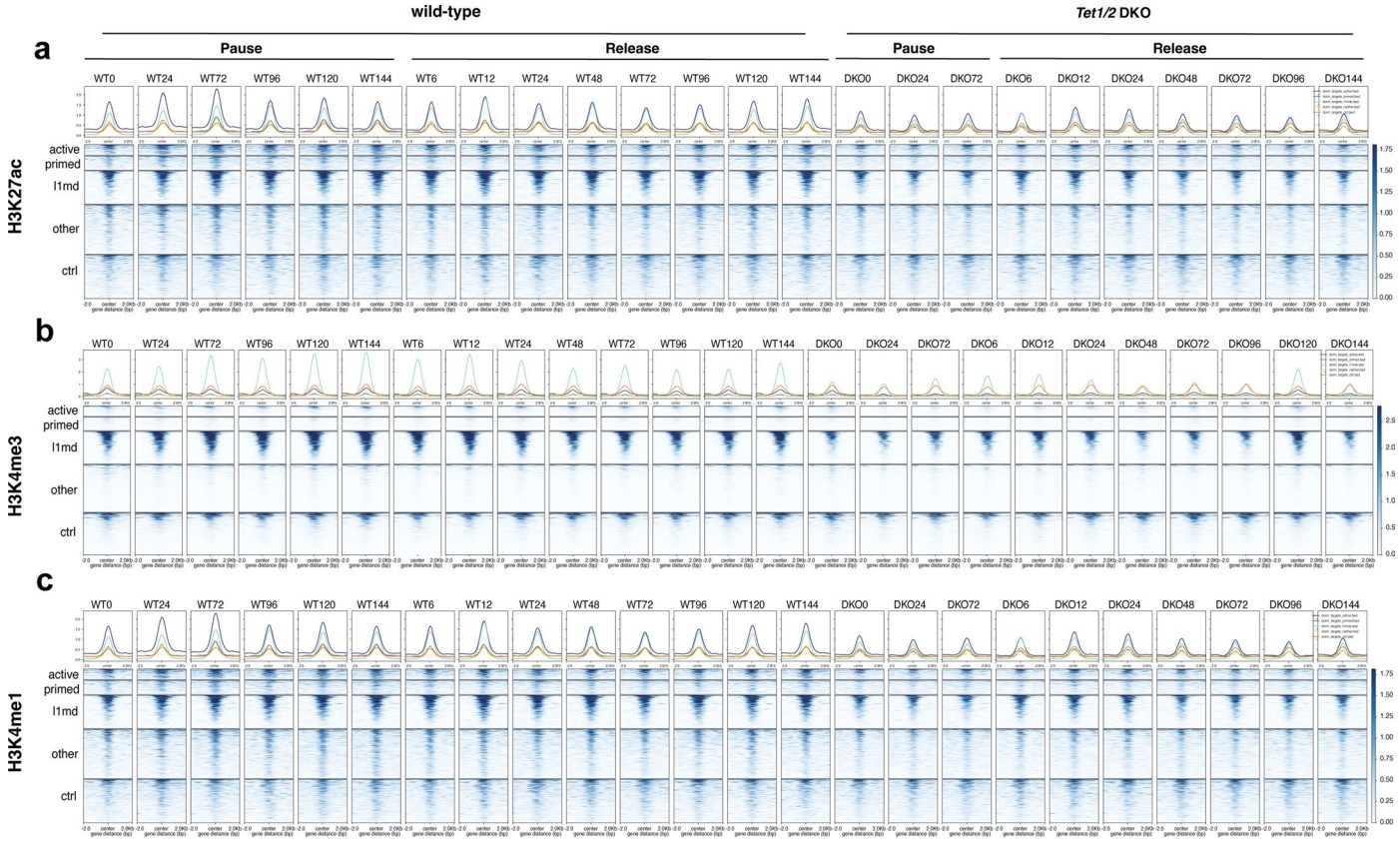

**Extended Data Fig. 9 | Histone marks at TET-dormancy target and control regions. a–c.** Distribution of H3K27ac (a), H3K4me3 (b), H3K4me1 (c) over pause and release time points in wild-type and Tet1/2 DKO cells. Plots are centered at peaks and 2 kb flanking regions on each side are shown. Histone marks show feature-appropriate patterns (for example no H3K4me3 at enhancers).

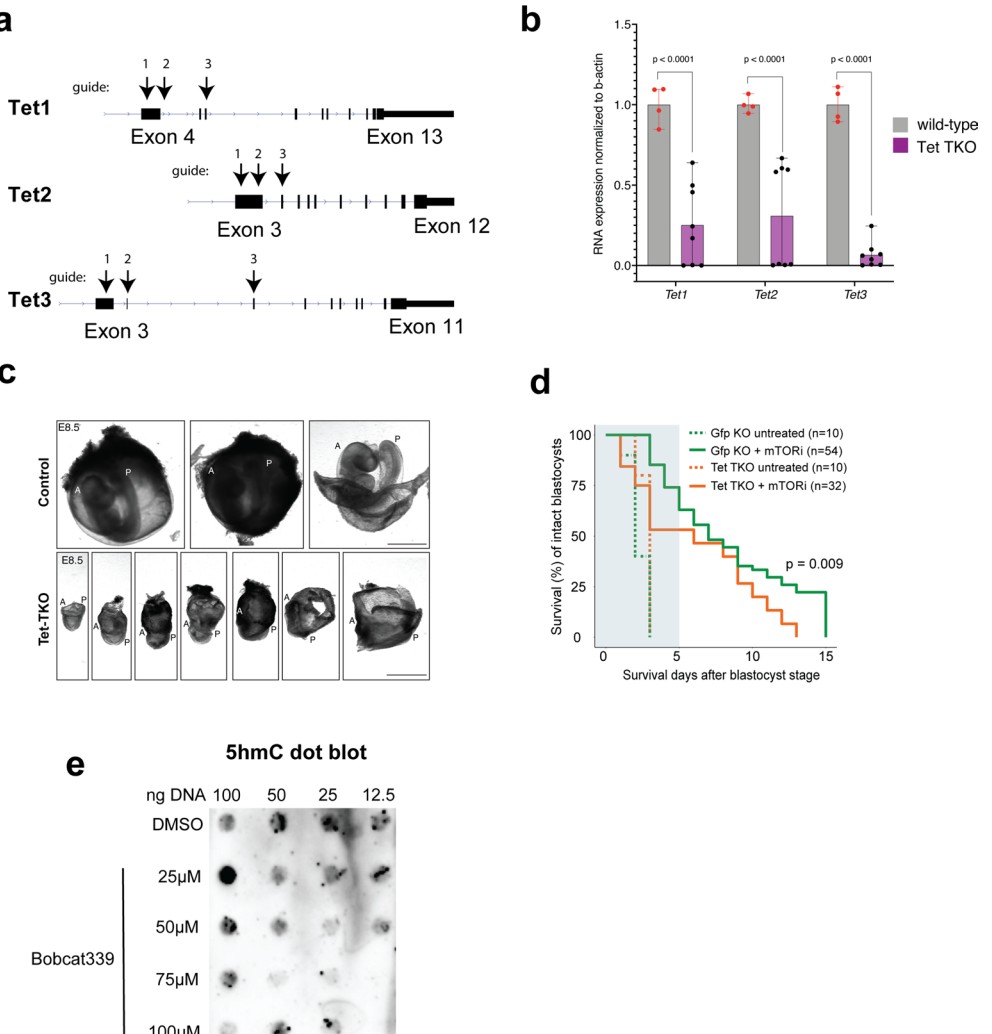

**5hmC dot blot**

**Extended Data Fig. 10 | Characterization of *Tet* KO embryos. a**. Schematics of Cas9-assisted *Tet* deletions via zygotic electroporation. **b**. RT-qPCR showing expression levels of Tet1/2/3 RNAs in control vs. targeted embryos. Each data point represents an embryo. 3 out of 8 embryos carry homozygous KOs for all *Tet* genes. Statistical test is a two-way ANOVA with Sidak-test for multiple comparison, comparing the means of *Tet*-expression of wild-type versus *Tet*-TKO embryos. Error bars represent the standard deviation. **c**. Phenotype of *Tet* TKO vs control embryos collected at E8.5 after retransfer. Scale bar = 500 μm. Three biological replicates were performed. A: anterior, P posterior. **d**. Survival curves of *Tet* TKO embryos under mTORi-induced dormancy conditions. Statistical test is log-rank test (R package survdiff) comparing Tet TKO with control under mTORi conditions. **e**. 5hmC dot blot for control and Bobcat339-treated ESCs.

# Reporting Summary

## Statistics

For all statistical analyses, confirm that the following items are present in the figure legend, table legend, main text, or Methods section.

| n/a | Confirmed | |
|---|---|---|
| ☐ | ☒ | The exact sample size (*n*) for each experimental group/condition, given as a discrete number and unit of measurement |
| ☐ | ☒ | A statement on whether measurements were taken from distinct samples or whether the same sample was measured repeatedly |
| ☐ | ☒ | The statistical test(s) used AND whether they are one- or two-sided *Only common tests should be described solely by name; describe more complex techniques in the Methods section.* |
| ☒ | ☐ | A description of all covariates tested |
| ☐ | ☒ | A description of any assumptions or corrections, such as tests of normality and adjustment for multiple comparisons |
| ☐ | ☒ | A full description of the statistical parameters including central tendency (e.g. means) or other basic estimates (e.g. regression coefficient) AND variation (e.g. standard deviation) or associated estimates of uncertainty (e.g. confidence intervals) |
| ☐ | ☒ | For null hypothesis testing, the test statistic (e.g. *F*, *t*, *r*) with confidence intervals, effect sizes, degrees of freedom and *P* value noted *Give P values as exact values whenever suitable.* |
| ☒ | ☐ | For Bayesian analysis, information on the choice of priors and Markov chain Monte Carlo settings |
| ☒ | ☐ | For hierarchical and complex designs, identification of the appropriate level for tests and full reporting of outcomes |
| ☐ | ☒ | Estimates of effect sizes (e.g. Cohen's *d*, Pearson's *r*), indicating how they were calculated |

*Our web collection on statistics for biologists contains articles on many of the points above.*

## Software and code

Policy information about availability of computer code

| Data collection | Mass spectrometry: MaxQuant software (v1.6.10.43)<br>Imaging: Zen black and Zen blue software (version 2.3)<br>Flow cytometry: BD FACSDIVA Software v8.0.1 configuration 2B-5YG-3R-2UV-6V |
|---|---|
| Data analysis | Image analysis: CellProfiler v4.2.1<br>Plotting: GraphPad Prism v10, ggplot2 v3.3.5, survminer 0.4.9<br>R v4.1.0<br>TF footprinting: ATAC-seq peak regions were called by MACS2 (2.2.7.1) with 75-bp shift and 150-bp extension. Differential TF footprints inside these peak regions were identified by TOBIAS v0.12.11 using Homer motifs (v4.7, 8-25-2014).<br>MACS2 (2.1.2_dev)<br>BWA v0.7.17<br>cutadapt v2.4<br>GATK v4.1.4.1<br>STAR v2.7.5a<br>stringtie v2.0.6<br>BSMAP v2.90<br>samtools v1.10<br>MOABS v1.3.2<br>DEseq2 (version 1.38.2)<br>FlowJo (v10.8.2) |

For manuscripts utilizing custom algorithms or software that are central to the research but not yet described in published literature, software must be made available to editors and reviewers. We strongly encourage code deposition in a community repository (e.g. GitHub). See the Nature Portfolio guidelines for submitting code & software for further information.

## Data

Policy information about availability of data

All manuscripts must include a data availability statement. This statement should provide the following information, where applicable:
- Accession codes, unique identifiers, or web links for publicly available datasets
- A description of any restrictions on data availability
- For clinical datasets or third party data, please ensure that the statement adheres to our policy

All NGS data sets (WGBS, RNA-seq, ATAC-seq, CUT&TAG, FLASH) have been deposited on the GEO database: GSE221470 (https://www.ncbi.nlm.nih.gov/geo/query/acc.cgi?acc=GSE221470)

Proteomics data has been deposited to the ProteomeXchange Consortium (http://proteomecentral.proteomexchange.org) via the PRIDE partner repository with the dataset identifiers  PXD039056.

Mouse reference genome (mm10) was used for mapping of sequencing data.

# Field-specific reporting

Please select the one below that is the best fit for your research. If you are not sure, read the appropriate sections before making your selection.

☒ Life sciences ☐ Behavioural & social sciences ☐ Ecological, evolutionary & environmental sciences

For a reference copy of the document with all sections, see nature.com/documents/nr-reporting-summary-flat.pdf

# Life sciences study design

All studies must disclose on these points even when the disclosure is negative.

| | |
|---|---|
| Sample size | No sample size calculation was performed. All experiments were repeated at least twice. For quantifications of stainings, 150 to 2500 cells were used based on the availability of cell numbers in culture. No batch effects are observed and thus the data from biological replicates are combined. Use of embryos were minimized and a conventional sample size of 4-10 embryos are used for stainings. Survival analysis was performed on 50-100 embryos as performed previously in: van der Weijden, V.A., Stötzel, M., Iyer, D.P. et al. FOXO1-mediated lipid metabolism maintains mammalian embryos in dormancy. Nat Cell Biol 26, 181–193 (2024). https://doi.org/10.1038/s41556-023-01325-3 |
| Data exclusions | No data were excluded. |
| Replication | All experiments are repeated at least twice independently and always with reproducible results. |
| Randomization | Samples were not randomized. For each biological replicate, corresponding time point samples are always processed and analyzed in parallel. Randomization was not possible due to different proliferation rates of cells. |
| Blinding | Investigators were not blinded to group allocation. The phenotype of the samples is obvious between proliferating and mTORi treated samples. Since all analysis is done via objective quantifications using software, blinding is not required. |

# Reporting for specific materials, systems and methods

We require information from authors about some types of materials, experimental systems and methods used in many studies. Here, indicate whether each material, system or method listed is relevant to your study. If you are not sure if a list item applies to your research, read the appropriate section before selecting a response.

## Materials & experimental systems

| n/a | Involved in the study |
|---|---|
| ☐ | ☒ Antibodies |
| ☐ | ☒ Eukaryotic cell lines |
| ☒ | ☐ Palaeontology and archaeology |
| ☐ | ☒ Animals and other organisms |
| ☒ | ☐ Human research participants |
| ☒ | ☐ Clinical data |
| ☒ | ☐ Dual use research of concern |

## Methods

| n/a | Involved in the study |
|---|---|
| ☒ | ☐ ChIP-seq |
| ☐ | ☒ Flow cytometry |
| ☒ | ☐ MRI-based neuroimaging |

## Antibodies

| | |
|---|---|
| Antibodies used | mESC and embryo stainings: Cells were then stained with primary antibodies 5hmC (ActifMotif, 39769; 1:200) and 5mC (Diagenode, C15200003; 1:100) overnight at 4 degree celsius. |

The cells were washed thrice with wash buffer (PBS-T, 2% BSA) for 10 min.
donkey anti-rabbit AF647 (Thermo Fisher, A32795, 1:1000) and donkey anti-mouse AF488 (Thermo Fisher, A21202, 1:1000).
Western-blotting: Primaries: Flag antibody (Merck, F3165, 1:1000), TET1 (Novus Biologicals, NBP2-19290, 1:1000), TET2 (Cell Signaling Technology, 45010S, 1:1000). Secondaries: anti-rabbit (Thermo, 31460, 1:1000)
CUT&TAG: TET1 (Novus Biologicals, NBP2-19290, 1:100), TET2 (Cell Signaling Technology, 45010S, 1:50), TFE3 (Sigma HPA023881, 1:50), IgG (Abcam, ab46540, 1:100). Secondary: guinea pig α-rabbit antibody (ABIN101961, Antibodies online, 1:100)
Immunoprecipitation: Flag antibody (Merck, F3165, 8µg)
For flowcytometry Alexa488-SSEA1 antibody (Biolegend, 125610,1:1000) or Alexa647-AnnexinV antibody (Invitrogen, A23204,1:250) was used.

Validation

Validation statements by manufacturer:
5hmC (ActifMotif, 39769) Applications Validated by Active Motif: MeDIP: 0.1 - 0.5 µl per IP, DB: 1:10,000 dilution.
5mC (Diagenode, C15200003) Validated by DotBlot and IF.
Flag antibody (Merck, F3165, 1:1000) IP, IF, WesternBlot
TET1 (Novus Biologicals, NBP2-19290) Use in Flow Cytometry reported in scientific literature (PMID:34246869). IHC-P, ChIP assay-Assay dependent.
TET2 (Cell Signaling Technology, 45010S) WB-Western Blot, IP-Immunoprecipitation, IHC-Immunohistochemistry, ChIP-Chromatin Immunoprecipitation, C&R-CUT&RUN, C&T-CUT&Tag, DB-Dot Blot, eCLIP-eCLIP, IF-Immunofluorescence, F-Flow Cytometry
TFE3 (Sigma HPA023881), RNAi knockdown, IF
Alexa488-SSEA1 antibody (Biolegend, 125610) immunofluorescent staining with flow cytometric analysis.
Alexa647-AnnexinV antibody (Invitrogen, A23204) Cell Viability, Proliferation & FunctionCellular ImagingFlow Cytometry.
donkey anti-rabbit AF647 (Thermo Fisher, A32795) (Advanced Verification This Antibody was verified by Relative expression to ensure that the antibody binds to the antigen stated.)
donkey anti-mouse AF488 (Thermo Fisher, A21202)(Mouse IgG (H+L) Highly Cross-Adsorbed Secondary Antibody (A-21202) in ICC/IF)
As indicated by the Nature protocol article from the Henikoff laboratory (doi: 10.1038/s41596-020-0373-x, "Unlike ChIP-seq, in which antibodies bind their epitopes in solution, CUT&RUN and CUT&Tag bind chromatin targets in situ. Therefore, we expect that antibodies successfully tested for specificity by immunofluorescence (IF) are likely to work.") Antibodies were validated via IF. Further every CUT&Tag experiments were performed with appropriate IgG controls to ensure specific cleavage.

# Eukaryotic cell lines

Policy information about cell lines

Cell line source(s)

Mouse E14 cells: Sarah Kinkley Lab at the Max Planck Institute for Molecular Genetics
Wild-type KH2, Dnmt3a/b DKO: Alexander Meissner Lab
Tet1/2/3 TKO ESCs: Jacob Hanna Lab
HEK239T cells: Denes Hnisz Lab

Authentication

The cell lines were not authenticated.

Mycoplasma contamination

Cell lines tested negative for mycoplasma in regular tests.

Commonly misidentified lines
(See ICLAC register)

No commonly misidentified lines were used.

# Animals and other organisms

Policy information about studies involving animals; ARRIVE guidelines recommended for reporting animal research

Laboratory animals

In vitro-fertilized (IVF) zygotes were electroporated to generate KOs, as previously described88. In brief, oocytes from superovulated B6D2F1 female mice (7-9 weeks old; Envigo) and sperm from F1B6xCAST were incubated for IVF, as previously described89. Pronuclei stage 3 (PN3) zygotes were rinsed with M2 (Sigma) and OptimMEM I (Gibco, 31985062) medium before electroporation 3 gRNAs per gene were designed targeting the first few exons. gRNAs were assembled with CAS9 into RNPs88. Embryos were electroporated on a NEPA21 (NEPAGENE, NEPA21) in a chamber with 5 mm electrode gap, and the following settings: 4 poring pulses with a voltage of 225 V, pulse length of 2 ms, pulse interval of 50 ms, decay rate of 10%, and uniform polarity, followed by 5 transfer pulses with a voltage of 20 V, pulse length of 50 ms, pulse interval of 50 ms, decay rate of 40%, and alternating polarity. Electroporated zygotes were rinsed in KSOM drops (Merck, MR-106-D) and cultured until blastocyst stage. In the case of embryo transfer, 15 blastocysts were transferred into each uterine horn of pseudopregnant female CD-1 (21-25 g, Envigo, age 7-12 weeks mice 2.5 days post-coitum (dpc). E8.5 stage embryos were isolated from the uteri of foster mice. The embryos were dissected in 1X HBSS (Gibco) on ice after the decicuda were removed. Embryos were washed in 1X PBS (Gibco) with 0.4% BSA and imaged on an Axiozoom (ZEISS) microscope. Images were processed with Fiji.
For KO validation qPCRs, RNA was isolated using Arcturus Pico Pure RNA isolation kit (Biosystems) and reverse transcribed using High-Capacity cDNA synthesis kit (KAPA biosystems). RT-qPCR was done using Kapa SYBR 2x master mix. B-actin was used for normalization. qPCR results were visualized using GraphPad Prism v10.
In vitro diapause
ESCs or embryos were treated with the mTOR inhibitor INK128 or RapaLink-1 at 200 nM final concentration for the durations specified in individual experiments. To obtain embryos, 10- to 12-week-old b6d2F1 mice were superovulated via intraperitoneal injection with PMSG (5IU/100 µl) on day 0,with HCG 5 IU/100µl on day 2, and sacrificed on day 3. Oocytes were collected and incubated with 10 µl of motile sperm in CARD Medium (CosmoBio, KYD-003-EX) for in vitro fertilization. After overnight culture, 2-cell stage embryos were transferred to a fresh drop of KSOM (Merck, MR-107-D) and cultured until the blastocyst stage.
Animal experimentation
Animal experiments were performed according to local animal welfare laws and approved by authorities (Landesamt für Gesundheit

und Soziales), covered by LaGeSo licenses ZH120, G0284/18, G021/19, and G0243/18-SGr1_G. Mice (7- to 12-week-old) were housed with enrichment material in ventilated cages (humidity 45-65%, temperature 20-24oC) on a 12h light/dark cycle and fed ad libitum.

| Wild animals | No wild animals were used. |
| Field-collected samples | No samples were collected from the field. |
| Ethics oversight | All animal experiments were performed according to local animal welfare laws and approved by local authorities (Landesamt für Gesundheit und Soziales), covered by LaGeSo licenses ZH120, G0284/18, G021/19, and G0243/18-SGr1_G. Mice were housed in ventilated cages and fed ad libitum. |

Note that full information on the approval of the study protocol must also be provided in the manuscript.

# Flow Cytometry

## Plots

Confirm that:

☒ The axis labels state the marker and fluorochrome used (e.g. CD4-FITC).

☒ The axis scales are clearly visible. Include numbers along axes only for bottom left plot of group (a 'group' is an analysis of identical markers).

☒ All plots are contour plots with outliers or pseudocolor plots.

☒ A numerical value for number of cells or percentage (with statistics) is provided.

## Methodology

| Sample preparation | Cells were dissocieted from plates, washed and resuspended in PBS containing 2% BSA and 5mM EDTA and kept on ice until sorting. |
| Instrument | FACS AriaFusion cell cytometer was used for analysis. |
| Software | Data were collected using BD FACSDIVA Software v8.0.1 and analyzed using FlowJo (v10.8.2). |
| Cell population abundance | Viable, single cells comprised ~90% of cells were sorted as input for RNAseq., ATAseq., and WGBS. |
| Gating strategy | FSC and SSC were used to set the first three gates to separate duplets from singlets by selecting the main population of cells and avoiding cell debris. |

☒ Tick this box to confirm that a figure exemplifying the gating strategy is provided in the Supplementary Information.

