## [Peer Review File · Nature Structural & Molecular Biology]

Peer Review Information

Manuscript Title: TET activity safeguards pluripotency throughout embryonic dormancy

Corresponding author name(s): Aydan Bulut-Karslioglu

Editorial Notes:

Redactions – transferred manuscripts (mention of the other journal) This manuscript has been previously reviewed at another journal. This document only contains reviewer comments, rebuttal and decision letters for versions considered at Nature Structural & Molecular Biology. Mentions of the other journal have been redacted.

Reviewer Comments & Decisions:

Decision Letter, initial version:

Message: Our ref: NSMB-A48795-T

15th Feb 2024

Dear Dr. Bulut-Karslioglu,

Thank you for submitting your manuscript "TET activity safeguards pluripotency throughout embryonic dormancy" (NSMB-A48795-T). We have now had the opportunity to discuss the paper and your response to the remaining referee comments. I am happy to let you know that the editorial team has agreed to, in principle, publish it in Nature Structural & Molecular Biology, pending minor revisions to address the referees' final requests, and to comply with our editorial and formatting guidelines.

To facilitate our work at this stage, it is important that we have a copy of the main text as a word file. If you could please send along a word version of this file as soon as possible, we would greatly appreciate it; please make sure to copy the NSMB account (cc'ed above).

Sincerely,

Carolina Perdigoto, PhD
Chief Editor
Nature Structural & Molecular Biology
orcid.org/0000-0002-5783-7106

Author Rebuttal to Initial comments

Reviewer #1 (Remarks to the Author):

The authors extensively addressed our comments, performing supplementary experiments to study the reactivation potential of the enhancers, providing an improved data representation to better contrast quantitative changes, and revising both the title and the text to put more focus on the role of TET-mediated DNA methylation in the diapause state rather than on the reactivation process. The data are analyzed and interpreted carefully, and the representation is clear. The conclusions are supported by their experiment. The results elucidate the TET-mediated DNA methylation role in modulating gene expression in the context of embryonal diapause, increasing our understanding of how pluripotency genes can maintain their epigenetic status during a transcriptional silent phase such as embryonic diapause. **In general, this study can be considered an important step to improve our understanding of the control of pluripotency and dormancy, both potentially useful to elucidate mechanisms taking place during development.** I would support publication upon correcting and improving data representation:

We thank the Reviewer for their time and valuable feedback, which greatly improved the paper.

- Figure 3C. The use of a PCA as unbiased representation of the data is appropriate here. It is unclear how the arrows overlayed on the PCA were created. While I understand that the authors want to highlight the trajectory, I would suggest to use a more appropriate data representation as for instance using color gradient to label the data points to illustrate the time course.

We did want to highlight the trajectory with the arrows, as the Reviewer points out. Even though we do find the arrows very useful to distinguish the trajectories of wild-type and *Tet* DKO cells from each other, we can remove these and try to make the point via color gradients.

- Figure 7a. The authors generated new time course datasets of histone marks to better characterise the process. This is relevant to the story. I however find the data representation and the text description very hard to read and understand. I would suggest to improve this in the final version of the manuscript.

We apologize for the confusion. This is a complex dataset which contains two time courses (in and out of dormancy) and two genotypes. We further streamlined the data representation and the corresponding text.

Reviewer #2 (Remarks to the Author):

The authors have addressed some of our questions, but some of the major concerns, particularly regarding the mechanism and in vivo relevance are not addressed. In my opinion, the current version does not meet ██████████ standard.

We appreciate the Reviewer's feedback, which helped improve our manuscript. However, we respectfully disagree with the statement that the Reviewer's concerns were not addressed. Below we reiterate how we addressed the Reviewer's concerns to the extent that is technically possible and ethically justifiable. Further below, we provide a point-by-point response to the remaining concerns.

1. The Reviewer had requested to validate the requirement for Tet activity in in vivo diapause. The word-for-word request was "To claim that TET proteins have a role in embryonic dormancy, the author need to provide in vivo evidence based on hormones-induced embryonic diapause". For this, we performed the technically highly challenging experiment of generating KOs, transferring these into recipient females while at the same time removing the ovaries surgically to induce (hormonally induced) in vivo diapause. These experiments showed, as the Reviewer asked, that TET activity is also required for in vivo diapause.
2. The Reviewer asked for a rescue experiment to validate that the defect in Tet DKO cells is due to DNA methylation. In addition to the rescue experiments we provided in the first submission, we performed additional rescue experiments by subjecting Dnmt TKO cells to TET inhibitors. The phenotype was once again rescued (Figure 2e).

3. The Reviewer asked 'Does KO or KD of TFE3/ZFP57 in mESCs exhibit a similar phenotype as TET DKO?', which we addressed by generating TFE3 inducible KD cells and showing that these exhibit a similar phenotype as TET DKO cells.
4. Generating longer time points, performing apoptosis assays, new embryo stainings, new RNA-seq analysis of Dnmt TKO cells are among other experiments we performed to address the Reviewer's major and minor concerns.

1. With regard to the in vivo function of TET in dormancy (my major concerns #1 and 2), the data does not support the conclusion. The statement that "half of the examined embryos were homozygous KOs for all three TETs (line 403)" indicating that TET preoteins are still retained in the other half of embryos. However, no embryo can survive like wild-type. Is this due to off-target effect of sgRNA or Tets exhibit haploinsufficiency? It is also strengte that TetTKO (Fig. ex10d) survive longer than DKO (Fig. 8b). Since the authors performed two or three gene KO with multiple sgRNAs, they need to carefully exclude the off-target effect. The authors are suggested to confirm their KO efficiency by PCR using single blastomere (PMID: 37696947). Related to Fig. ex10b,c, phenotypes must be linked to the KO genotype. Also, the frequency of the phenotypes must be provided.

In this study we provided three different embryo survival datasets:

- 1) Generation of Tet1/2 DKO, then in vitro pause via mTORi,
- 2) Generation of Tet1/2 DKO, then in vivo diapause via transfer+ovariectomy.
- 3) Generation of Tet1/2/3 TKO, then in vitro pause via mTORi (this was the first dataset we had in the original submission).

The Reviewer asks to link the phenotype to the genotype in each embryo via single blastomere genotyping. We have previously successfully performed single-blastocyst genotyping, which barely provides enough material for the genotyping PCR. We did not find the single-blastomere genotyping protocol in the cited paper. In any case, even if single-blastomere genotyping were possible, it would not be feasible in our experiments, as we strive to subject 20-60 embryos to survival tests in each condition and each of these embryos would then require blastomere excision procedure. To assess the efficiency of the KOs, we chose to transfer a subset of the targeted embryos into female mice, and reisolate them after a few days. This approach provides ample material for genotyping, even though it is not matched individually to each embryo. In addition, we also assessed the development of targeted embryos and observed developmental delay as reported before^{1,2}.

The statement 'no embryo can survive like wild-type' is unfortunately not entirely correct. The survival curves represent compiled survival data from many embryos (n value indicated in each figure). Even in wild-type embryos, there is always variable response to mTORi, with some embryos collapsing earlier than others (the variable response also true for in vivo diapause³). Therefore, in the TKO vs control. setting, some TKO embryos survive longer than control embryos. What the Reviewer might have meant is that none of the TKO embryos survive longer than the maximum-surviving control embryo. This might possibly be explained by combined haploinsufficiency of the three TET genes. The variable response in both wild-type (or control) embryos and in TKO embryos may stem from the background genetic buildup as well. While the question of the source of this variability is worth exploring further, we believe it is beyond the scope of this study.

To support Tet's role in embryonic dormancy, the authors should at least perform DNA methylation analysis in embryo. WGBS or RRBS in epiblast has been performed before (PMID: 32064321; PMID: 29203909).

As mentioned above, we provide in vivo evidence for the requirement for Tet activity during embryonic diapause (current Figure 8e). Additionally, we do validate the increase in DNA methylation in diapaused embryos via DNA methylation (5mC) stainings and quantifications in embryos, including in vivo-diapaused ones (Figure 1d). Therefore both the phenomenon (DNA methylation increase in dormancy) and the regulator (Tets) have been validated. The WGBS data that is requested by the referee will show a global increase in DNA methylation in embryos, which is already obvious in Figure 1d. However, it is unclear whether it will have enough coverage/sensitivity to pick up differences at regulatory elements. In the papers cited by the Reviewer, hundreds of embryos per condition are used to generate WGBS datasets. RRBS would require fewer embryos but is not a viable option to address this question because the enhancers do not contain enough CGs to survive the

enzymatic digestions done during processing of RRBS samples, and are poorly represented in the final dataset (we tried it on ESCs and the dataset did not yield good results).

It is important to note that mTORi-treated (paused) ESCs have been shown to closely represent the gene expression profile of the diapaused epiblast⁴. We have previously also shown that functional mechanisms identified using paused ESCs reflect the corresponding phenomena in the in vivo-diapaused embryo⁵. Given that paused ESCs are indeed a faithful model of the diapaused epiblast and that we did support our results here with functional validation on embryos, we find it **ethically unjustifiable** to use hundreds of embryos to generate methylation datasets which may in the end not be sensitive enough to analyze TET-dormancy targets.

2. The authors need to provide more evidence for their TETi treatment, especially in embryos. Do they show loss of 5hmC in embryo? To evaluate side-effect, embryos needed to be treated over days. Also, the authors did not try to improve their dot-blot quality (Fig. ex10e) despite we requested a better quality.

Previously the Reviewer asked us to (further) complement the TET inhibitor experiments with genetic perturbation, which we did (now Figure 8e). Additionally, the Reviewer asked whether TETi compromises blastocyst formation rate, to which we replied to state that it does not. The Reviewer did not ask to stain embryos, and therefore we did not. To the point of dot-blot quality, it is known that dot blots are very variable and multiple iterations lead to cherry-picking instead of improving quality. Since we strongly complemented the TETi inhibitors with genetic perturbations, we believe there is sufficient evidence to suggest that TET activity is important during dormancy transition. However, we understand that the various perturbation approaches may make the paper complicated to understand. Therefore, we revised the text and figures to streamline this part and make it more accessible to readers.

For the statement of “modifications to embryo culture media based on discovered mechanisms of dormancy can improve the efficiency of in vitro developmental pausing. (line 428-429)” must be supported by statistic analysis.

The statistical analysis is already provided in Figure 8f.

Also, the authors need to consider batch effect giving mTORi resulted in over 75% survival (Fig. 8c), while mTORi+PBS treatment was much lower.

The Reviewer is correct that there are batch effects. These depend on the specific set of embryos, the inhibitor, the embryo generation method, etc. We are aware of the batch variation and for this reason include a matching control condition in all embryo experiments we perform.

3. For my major concern #4, some key analyses are still missing although they claim to observed correlations between transcriptome and Tet-targeted regions and TF binding.

We would like to clarify: There is no correlation between TET/TF-targets and transcriptome because these are transcriptionally inert changes. With regards to this point and also regarding chromatin accessibility, the Reviewer previously stated that these findings are against the general rules of gene regulation. This is true and is the very reason that makes the mechanism novel.

The authors need to provide transcriptome data on TFE3 KD or KO.

The reviewer previously asked us “Does KO or KD of TFE3/ZFP57 in mESCs exhibit a similar **phenotype** as TET DKO?”. We showed that indeed TFE3 KD exhibited the same phenotype, thereby addressing the concern. We were not asked to perform transcriptomics analysis on these cells. While the results will surely be valuable, this experiment goes deeper into the mechanism of TF-mediation of dormancy, which is one step removed from the original mechanism that we describe here in detail. The TF-related regulation will be explored in future studies.

These data are essential to demonstrate the role of the TET-TF axis on establishing transcriptionally inert chromatin. To clarify Tet-DKO defect in reactivation, it is better to perform omics-analysis at 48hr after induction.

We were not asked to provide this experiment during the revision. While a detailed investigation of reactivation dynamics is certainly worth performing, this request again is beyond the scope of the current study.

Also, the authors should explain what do they mean for “chromatin adaptation (line 388)”.

By ‘chromatin adaptation’ we meant all of the dynamic changes that occur during this cellular transition. Specifically, TET-mediated DNA demethylation and TF binding is meant.

4. Further analysis on TFE3 is needed. It is better to show by Western blotting for depletion efficiency.

We show by immunofluorescence the complete depletion of the protein in the KD cell line. TFE3 is a protein known to shuttle between the nucleus and the cytoplasm, and we used IF to both gauge expression levels and protein localization in intact cells. The Reviewer did not register a preference for validation method in their original Review.

TET-TFE3 interaction needs to be validated by Co-IP experiment.

This interaction is already validated by an IP-mass spec experiment (Figure 5f, EDF 6d).

mTORi 120h data and growth curve are needed to be compared with Tet-KO. It is better to provide SSEA-1 analysis and reactivating experiment to support TET-TFE3 axis. Also, TFE3’s role in dormancy need to be confirmed in embryo context.

The Reviewer asked us to assess the phenotype of TFE3 KD cells, which we have done. The above requests were not brought up by the Reviewer in their original report. We respectfully think it is not appropriate to demand these extensive new experiments as follow up to our successful addressing of the original concern. Additionally, while the results will surely be valuable, this experiment goes deeper into the mechanism of TF-mediation of dormancy, which is one step removed from the original mechanism that we describe here in detail. The TF-related regulation will be explored in future studies.

5. Other unaddressed questions include:

- 5mC is not observed in Fig.1d control embryo, which is inconsistent with previous knowledge (PMID: 21747414; 22262693). DAPI signal between control and treated embryos is different, thus, DAPI normalization is not suitable.

This is due to the laser settings. Since the diapaused embryos show much higher 5mC signal, the laser was adjusted so as to not saturate this signal. This is standard and customary practice in generating imaging datasets.

- Line119-122 authors provide new data of 5mC in reactivated diapaused embryos, which is comparable to E4.5. They should describe how to induce reactivation in method and confirm embryo reactivation. Notably, the in vivo reactivated embryos should be implanted to uterus after reactivation, thus this data is inconsistent with the knowledge that DNA methylation level is dramatically increased after implantation (PMID: 29203909; PMID: 31827285).

The embryos were reactivated in vitro in blastocyst culture media and were not transferred back into female mice. Thus, the embryos are still in the preimplantation stage. Therefore there is no inconsistency with existing literature, since the Reviewer refers to post-implantation embryos.

- It is not clear statistical analysis represent which data in Fig. 2e.

The statistical analysis refers to wild type+mTORi+TETi vs Dnmt TKO+mTORi+TETi dataset.

- Any correlation between gain of DNA methylation and ATAC-seq peak change, related with Fig. 1c, e and Fig. 3c, d?

We have not done this correlation because DNA methylation and ATAC-seq datasets almost exclusively probe different genomic regions. DNA methylation increases globally throughout the genome except promoters and TET-dormancy targets (which include a large number of enhancers). ATAC-seq provides info about accessible sites, which correspond mostly to promoters and enhancers. Therefore, we would not expect high correlation between increased DNA methylation and ATAC-seq data in wild-type cells.

• “muted (H3K4me1) or diminished (H3K27ac) response (line 364)” is not clear, need explanation. The logic “Since these events do not lead to increased expression of associated genes, we conclude that TET activity is required for transcriptionally inert chromatin adaptations to maintain pluripotency in the silent genome of dormant cells. (line 373-375)” is not clear.

We meant that, in DKO cells, the dynamic changes in the studied chromatin marks are either greatly reduced (for H3K4me1) or completely absent (for H3K27ac) (Figure 7a). As also requested by Reviewer 1, we will revise the text to improve clarity.

Reviewer #3 (Remarks to the Author):

This is still a very complex study on the control of dormancy using stem cells from mice as a model. The authors have provided a very extensive 26 page rebuttal and modified text with multiple clarifications and removal of many overstatements. A great many generalisations have now been clarified. While ESCs still may not represent an appropriate model for diapause, the additional data provided on in vivo experiments is somewhat reassuring.

We thank the Reviewer for their helpful feedback and appreciation of our study. We would like to further clarify the logic of using ESCs as part of the study:

Embryonic diapause refers to the dormant state of the entire embryo. The acute sensitivity of the embryo to its surroundings, i.e. sensing the absence of nutrients/growth factors and/or presence of inhibitors, underlies the ability to enter diapause. However, it is important to note that the different cell types within the blastocyst show distinct responses to the same triggers that induce diapause. For example, Kamemizu et al⁶ showed that the inner cell mass (ICM) ceases proliferation faster than the trophectoderm (TE) tissue. We have shown recently that the ICM and TE show distinct energy dynamics as well, with the ICM readily consuming cellular lipids and TE accumulating them⁵. We and others showed that the nuclei of dormant ICM cells gain large heterochromatin domains, which are absent from normal ICM cells^{5,7}. This is not the case for TE cells, likely because these already accumulate heterochromatin earlier (upon setting of the TE fate and separation from the ICM).

Importantly, specific cellular strategies appear to be necessary to maintain pluripotency of the epiblast (which is normally a transient state) during diapause. Jenny Nichols and colleagues showed decades ago that the LIF/gp130 pathway is upregulated in diapause and protects pluripotency⁸. Indeed, this finding explains why LIF is needed in stem cell cultures, even though it is dispensable in normal preimplantation development. More recently, Ivan Bedzhov and colleagues showed a similar requirement for Wnt pathway activity/Esrrb in the diapaused epiblast⁹. These findings point to not only an alternative pluripotency regulation in the diapaused epiblast, but also highlight the shared regulatory basis of the diapaused epiblast and stem cells in culture. Importantly, we have previously shown that the transcriptional signature of dormant ESCs closely resemble that of the diapaused epiblast, showing a high degree of molecular similarity in these two states^{4,10}.

In this paper we sought to identify chromatin mechanisms that safeguard pluripotency in dormant pluripotent cells, which carry increased DNA methylation and heterochromatin domains that are normally not found at this stage. Since our study is centered on pluripotency mechanisms, and due to the reasons outlined above, we believe that ESCs are a suitable model to study paused pluripotency. Nevertheless, we complemented the ESC data with molecular and functional experiments on embryos.

The additional bar graphs should not be included as they are n=2.

It is unfortunately unclear to us which bar graphs are meant. We now complied with all editorial requests, which includes revision of graphs in cases where two biological replicates were performed.

References

1. Dawlaty, M. M. *et al.* Combined Deficiency of Tet1 and Tet2 Causes Epigenetic Abnormalities but Is Compatible with Postnatal Development. *Dev Cell* **24**, 310–323 (2013).
2. Dawlaty, M. M. *et al.* Loss of Tet Enzymes Compromises Proper Differentiation of Embryonic Stem Cells. *Dev Cell* **29**, 102–111 (2014).
3. Battle-Morera, L., Smith, A. & Nichols, J. Parameters influencing derivation of embryonic stem cells from murine embryos. *Genesis* **46**, 758–767 (2008).
4. Bulut-Karslioglu, A. *et al.* Inhibition of mTOR induces a paused pluripotent state. *Nature* **540**, 119–123 (2016).
5. Weijden, V. A. van der *et al.* FOXO1-mediated lipid metabolism maintains mammalian embryos in dormancy. *Nat. Cell Biol.* 1–13 (2024) doi:10.1038/s41556-023-01325-3.
6. Kamemizu, C. & Fujimori, T. Distinct dormancy progression depending on embryonic regions during mouse embryonic diapause†. *Biol. Reprod.* **100**, 1204–1214 (2019).
7. Fu, Z. *et al.* Integral Proteomic Analysis of Blastocysts Reveals Key Molecular Machinery Governing Embryonic Diapause and Reactivation for Implantation in Mice1. *Biol. Reprod.* **90**, Article 52, 1-11 (2014).
8. Nichols, J., Chambers, I., Taga, T. & Smith, A. Physiological rationale for responsiveness of mouse embryonic stem cells to gp130 cytokines. *Development* **128**, 2333–2339 (2001).
9. Fan, R. *et al.* Wnt/Beta-catenin/Esrrb signalling controls the tissue-scale reorganization and maintenance of the pluripotent lineage during murine embryonic diapause. *Nat Commun* **11**, 5499 (2020).
10. Boroviak, T. *et al.* Lineage-Specific Profiling Delineates the Emergence and Progression of Naive Pluripotency in Mammalian Embryogenesis. *Dev. Cell* **35**, 366–382 (2015).

Final Decision Letter:

Message: 10th Apr 2024

Dear Dr. Bulut-Karslioglu,

We are now happy to accept your revised paper "TET activity safeguards pluripotency throughout embryonic dormancy" for publication as an Article in Nature Structural & Molecular Biology.

Your paper will be published online soon after we receive proof corrections and will appear in print in the next available issue. You can find out your date of online publication by contacting the production team shortly after sending your proof corrections.

You may wish to make your media relations office aware of your accepted publication, in

case they consider it appropriate to organize some internal or external publicity. Once your paper has been scheduled you will receive an email confirming the publication details. This is normally 3-4 working days in advance of publication. If you need additional notice of the date and time of publication, please let the production team know when you receive the proof of your article to ensure there is sufficient time to coordinate. Further information on our embargo policies can be found here: <https://www.nature.com/authors/policies/embargo.html>

If you have not already done so, we strongly recommend that you upload the step-by-step protocols used in this manuscript to the Protocol Exchange. Protocol Exchange is an open online resource that allows researchers to share their detailed experimental know-how. All uploaded protocols are made freely available, assigned DOIs for ease of citation and fully searchable through [nature.com](https://www.nature.com). Protocols can be linked to any publications in which they are used and will be linked to from your article. You can also establish a dedicated page to collect all your lab Protocols. By uploading your Protocols to Protocol Exchange, you are enabling researchers to more readily reproduce or adapt the methodology you use, as well as increasing the visibility of your protocols and papers. Upload your Protocols at www.nature.com/protocolexchange/. Further information can be found at www.nature.com/protocolexchange/about.

Please note that *Nature Structural & Molecular Biology* is a Transformative Journal (TJ). Authors may publish their research with us through the traditional subscription access route or make their paper immediately open access through payment of an article-processing charge (APC). Authors will not be required to make a final decision about access to their article until it has been accepted. Find out more about Transformative Journals

You will not receive your proofs until the publishing agreement has been received through

our system.

Sincerely,

Dimitris Typas
Associate Editor
Nature Structural & Molecular Biology
ORCID: 0000-0002-8737-1319